# A TALE/HOX code unlocks WNT signalling response towards paraxial mesoderm

Luca Mariani[1,3], Xiaogang Guo[1,3], Niels Alvaro Menezes[1], Anna Maria Drozd [1], Selgin Deniz Çakal [1], Qinhu Wang [2] & Elisabetta Ferretti[1✉]

One fundamental yet unresolved question in biology remains how cells interpret the same signalling cues in a context-dependent manner resulting in lineage specification. A key step for decoding signalling cues is the establishment of a permissive chromatin environment at lineage-specific genes triggering transcriptional responses to inductive signals. For instance, bipotent neuromesodermal progenitors (NMPs) are equipped with a WNT-decoding module, which relies on TCFs/LEF activity to sustain both NMP expansion and paraxial mesoderm differentiation. However, how WNT signalling activates lineage specific genes in a temporal manner remains unclear. Here, we demonstrate that paraxial mesoderm induction relies on the TALE/HOX combinatorial activity that simultaneously represses NMP genes and activates the differentiation program. We identify the BRACHYURY-TALE/HOX code that destabilizes the nucleosomes at WNT-responsive regions and establishes the permissive chromatin landscape for de novo recruitment of the WNT-effector LEF1, unlocking the WNT-mediated transcriptional program that drives NMPs towards the paraxial mesodermal fate.

[1] Novo Nordisk Foundation Center for Stem Cell Biology (DanStem), University of Copenhagen, Copenhagen, Denmark. [2] State Key Laboratory of Crop Stress Biology for Arid Areas and College of Plant Protection, Northwest A&F University, Yangling, Shaanxi, China. [3]These authors contributed equally: Luca Mariani, Xiaogang Guo. ✉email: elisabetta.ferretti@sund.ku.dk

In mammals, skeletal muscles and vertebrae originate from the paraxial mesoderm, which includes unsegmented pre-somitic mesoderm (PSM) and somites[1–3]. PSM formation, which represents the earliest event of musculoskeletal development, relies on the availability of expanding neuromesodermal progenitors (NMPs) that also contribute to the spinal cord[3–6]. NMPs reside in the caudal lateral epiblast (CLE) and tailbud, from where they sustain axial elongation[4,7,8]. Intricate molecular circuits regulated by WNT signalling and involving the transcription factors (TFs) HOX, CDX, SOX2, BRACHYURY (T-BRA), TBX6 and MSGN1, balance NMP expansion and differentiation[9–17]. NMPs can adopt neural fate by downregulating *T-Bra* and maintaining *Sox2* expression. Conversely, NMPs acquiring PSM fate retain *T-Bra* expression, downregulate *Sox2* and upregulate *Tbx6* and *Msgn1*, transitioning into nascent mesodermal progenitor cells (MPCs) that fuel PSM development[14,18]. Ultimately, somite formation depends on the speed by which NMPs leave the CLE to acquire paraxial mesoderm fate[1].

HOX TFs have been historically implicated in axial elongation, with their sequential activation playing a fundamental role in timing PSM development[16,19,20]. Posterior *Hox* genes *(Hox10-13)* reduce mesoderm ingression by repressing WNT and are responsible for body axis termination[11,21]. Conversely, the anterior *Hox* genes *(Hox1-8)* control paraxial mesoderm formation by modulating cell ingression into the primitive streak (PS)[19]. Nevertheless, the mechanism by which *Hox* genes promote PSM specification remains largely unknown. TFs belonging to the TALE (three-amino acid loop extension) family like the PBX and MEIS factors have distinct ability to interact with HOX proteins and provide HOX with DNA-binding specificity[22]. Given that PBX proteins are obligate anterior HOX cofactors, they represent prominent candidates for controlling the distinct response to individual HOX factors[23]. PBX1, PBX2 and PBX3 play fundamental roles during organogenesis[22,24,25]. Compound deletion of PBX1 and PBX2 (hereafter PBX) uncovers a crucial role for these factors in axial/appendicular patterning[24,25], a surprisingly late phenotype as they are widely expressed at earlier embryonic stages[26]. Importantly, their role in mesodermal lineage specification remains elusive. Given the broad expression of PBX, it is still unclear how cooperative interactions with HOX provide lineage specificity.

By employing in vitro differentiation systems and in vivo mouse models, we show that the patterning abnormalities in *Pbx1/2* double mutant embryos (hereafter *Pbx*) can be traced back to the PSM specification stage. In *Pbx* mutants, NMPs exhibit reduced ability to generate paraxial mesoderm, being unable to migrate, remaining trapped in the tailbud in a NMP/MPC transition state. Using single-cell sequencing, chromatin accessibility analyses and iterative rounds of DNA:protein interaction assays, we demonstrate that TALE/HOX proteins generate the DNA-binding context for the recruitment of the WNT-effector LEF1, and consequently promote the expression of PSM genes, including the master regulator *Mesogenin1 (Msgn1)*[10,15]. Ultimately, the activation of the paraxial mesoderm gene regulatory networks (GRNs) rely on the activity of a WNT-HOX integrated code, which drives a multistep process involving changes in the chromatin landscape at the WNT-responsive elements supporting the shift from NMP expansion to paraxial mesoderm differentiation. Thus, TALE/HOX act as a molecular switch that triggers alternative WNT signalling cellular responses, unlocking PSM genes and promoting somite formation.

## Results

### PBX proteins control the formation of the nascent paraxial mesoderm.
Addressing how *Hox* genes influence WNT signalling is critical for paraxial mesoderm formation and somitogenesis[11,19]. The

HOX cofactors PBX are broadly expressed in the tailbud of embryonic day (E) 8.5 mouse embryos, including PS and NMPs (Supplementary Fig. 1a), questioning their function in the early stages of somitogenesis. To dissect the role of PBX at early stages, we inactivated *Pbx1* in *Pbx2* mutant mice using the *Sox2^Cre* allele to obtain *Pbx1^−/−;Pbx2^−/−* double-knockout (*Pbx-DKO*) and *Pbx1^−/−;Pbx2^+/−* compound mutant (*Pbx-com*) embryos. *Pbx-DKO* display shortened trunk, enlarged tailbud and rudimentary somites (Fig. 1a and Supplementary Fig. 1b). Furthermore, FOXC2 and UNCX4.1 immunofluorescence (IF) revealed abnormal posterior somite morphogenesis in E8.5 *Pbx-DKO* and *Pbx-com* (Fig. 1b and Supplementary Fig. 1c). To pinpoint the role of PBX proteins in PSM development (Fig. 1c), we performed single-cell RNA sequencing (scRNA-seq) on cells isolated from the tailbuds of control and *Pbx* mutants at E8.5 and E9.0 (Fig. 1d and Supplementary Data 1). After stringent quality filtering, batch-effect correction and data integration of all sequenced embryos, unsupervised clustering analyses revealed 13 different populations, including NMPs, MPCs and PSM (Fig. 1e, Supplementary Fig. 1d, e and Supplementary Data 2). Each embryo contributed to every cluster, irrespective of the genotype. However, controls and mutants exhibited markedly different proportions of cell types populating each cluster. Specifically, *Pbx-DKO* and *Pbx-com* mutant cells were overrepresented in the posterior PSM cluster, while being largely reduced in anterior PSM, suggesting a block on PSM maturation. Quantification of cluster composition confirmed that cells displaying MPC features (T-Bra^pos;Tbx6^pos) significantly accumulate in the posterior PSM cluster of *Pbx-DKO* and *Pbx-com* at the expense of anterior PSM (Fig. 1f). Specifically focussing on NMPs, MPCs and PSM cell populations revealed that the *Pbx-DKO* and *Pbx-com* NMPs and MPCs/pPSM clusters included cells that exhibited a mixed NMP/MPC profile, expressing high levels of *T-Bra*, *Sox2*, *Wnt3a*, *Tbx6* and 5′ *Hox* genes (Fig. 1g and Supplementary Fig. 1f–h). To understand the transition of NMPs towards PSM, we applied pseudotemporal ordering analyses and confirmed defective paraxial mesoderm differentiation in *Pbx-DKO*, resulting in inflation of the progenitor reservoir at the expense of mature PSM (Fig. 1h and Supplementary Fig. 1i).

The impairment of PSM formation and the increased number of MPCs with NMP features in *Pbx-DKO* and *Pbx-com* prompted us to address whether the spatial organisation of the progenitor zone was affected[27]. By IF, we observed an expansion of T-BRA^pos;SOX2^pos NMPs in the *Pbx-DKO* and *Pbx-com* tailbuds (Fig. 2a and Supplementary Fig. 2a). By contrast, the mesodermal marker TBX6 was strongly downregulated in the PSM and upregulated in the CLE of *Pbx-DKO* and *Pbx-com* mutants (Fig. 2a), confirming the accumulation of aberrant progenitors observed by scRNA-seq. The distribution of progenitor populations assessed in the caudal and medial parts of the CLE, where cells preferentially adopt a PSM fate, revealed that in *Pbx-DKO* and *Pbx-com* the number of NMPs (T-BRA^pos;SOX2^pos) and MPCs (T-BRA^pos;TBX6^pos) increased, while the amount of mature PSM (TBX6^pos) cells was dramatically reduced (Fig. 2b, c and Supplementary Fig. 2b). Notably, the proportion of T-BRA^pos;TBX6^pos cells relative to TBX6^pos cells was significantly enhanced in *Pbx-DKO* and *Pbx-com*, suggesting an impaired maturation of MPCs (Fig. 2c). Furthermore, NMPs robustly accumulate in the mutant posterior tailbuds, as opposed to the fewer NMPs observed in controls (Fig. 2d). Thus, the analysis of the progenitor allocation in *Pbx* mutant embryos corroborated the scRNA-seq findings (Fig. 1e–h), and provided evidence that *Pbx-DKO* and *Pbx-com* NMPs are hindered in their progress towards PSM, remaining trapped in the CLE in a NMP/MPC state.

Given that WNT3A supports both NMP maintenance and PSM differentiation[14,18] (Supplementary Fig. 2c), we tested whether the PSM defects in the *Pbx* mutants were the consequence of perturbed

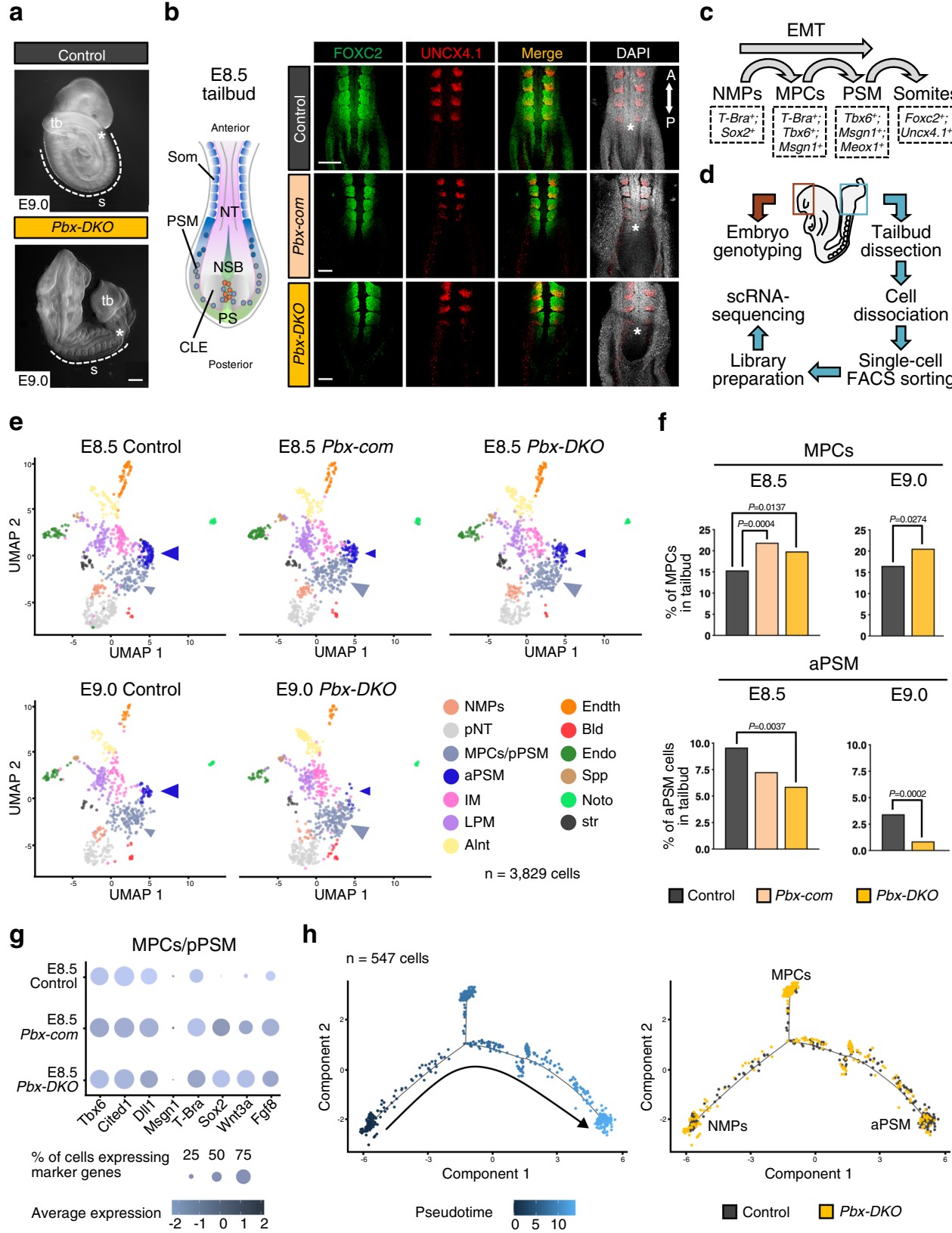

WNT expression, and found that *Wnt3a* expression domain was increased in the *Pbx-com* tailbud (Supplementary Fig. 2d), in agreement with the observed progenitor zone expansion (Fig. 2a, d). Therefore, although WNT signalling is a major mediator of PSM specification, the phenotypes of *Pbx* mutants are not associated with

the depletion of the *Wnt3a* ligand, suggesting that PBX could have a role in modulating how NMPs respond to this pathway. Remarkably, *Pbx* mutants have an expanded NMP population, implicating PBX in the regulation of the WNT-dependent NMP-to-PSM transition.

**Fig. 1 PBX proteins control the transition of NMPs towards PSM, underlying the acquisition of paraxial mesoderm identity. a** Gross morphology of E9.0 *Pbx1*[−/−]*;Pbx2*[−/−] (*Pbx-DKO*) embryos showing enlarged tailbud (tb), shortened trunk and rudimentary somites (s). Scale bar: 200 μm. **b** Dorsal views of E8.5 *Pbx1*[−/−]*;Pbx2*[+/−] (*Pbx-com*) and *Pbx-DKO* mouse embryos stained with FOXC2 (green) and UNCX4.1 (red) antibodies display abnormal somitogenesis. The cartoon on the left is a schematic representation of the E8.5 mouse tailbud. CLE caudal lateral epiblast, NSB node-streak border, PS primitive streak, PSM pre-somitic mesoderm, Som somites, NT neural tube, A anterior, P posterior. Scale bars: 70 μm. In **a** and **b**, asterisks denote the newly generated somites. **c** Schematics illustrating paraxial mesoderm differentiation from bipotent NMPs to somites, showing intermediate states and associated lineage markers. EMT epithelial-to-mesenchymal transition. **d** Strategy used to perform single-cell RNA-seq (scRNA-seq) of mouse embryonic tailbuds. **e** ScRNA-seq of E8.5/E9.0 control, *Pbx-com* and *Pbx-DKO* embryonic tailbuds. Unsupervised clustering and UMAP visualisation reveal reduced aPSM (dark blue arrowhead) and expanded MPCs/pPSM (light blue arrowhead) cell populations in *Pbx* mutants. Individual cells are coloured according to annotated cluster identities: NMPs neuromesodermal progenitors, pNT pre-neural tube, MPCs/pPSM mesodermal progenitor cells/posterior pre-somitic mesoderm, aPSM anterior pre-somitic mesoderm, IM intermediate mesoderm, LPM lateral plate mesoderm, Alnt allantois, Endth endothelial cells, Bld blood, Endo endoderm, Spp splanchnopleura, Noto notochord, str stressed cells. **f** Percentage of MPCs and aPSM cells in control (black), *Pbx-com* (light pink) and *Pbx-DKO* (yellow) tailbuds at E8.5 and E9.0. *P*-values indicated in the figure were calculated by two-sided Fisher's exact test. **g** Expression analysis of selected lineage markers within the MPCs/pPSM cluster isolated from control and *Pbx* mutant tailbuds. Dot plot shows the relative expression of the NMP markers *T-Bra*, *Sox2*, *Wnt3a*, *Fgf8*, and the mesodermal genes *Tbx6*, *Cited1*, *Dll1*, *Msgn1*, indicating co-expression of NMP and MPC markers in *Pbx-com* and *Pbx-DKO* embryos. Colour bar indicates the intensity associated with normalised expression values. **h** Monocle's pseudotemporal ordering of paraxial mesoderm-related clusters displaying the progressive maturation of control cells (black) from NMPs to aPSM and the defective NMP-to-PSM transition of *Pbx-DKO* cells (yellow).

**PBX proteins promote the transition from NMPs towards PSM**. To address PBX function during PSM differentiation, we took advantage of our recently developed in vitro differentiation protocol (Mariani, L. et al., in preparation[28]) that efficiently drives mouse epiblast stem cells (EpiSCs) towards PSM (Fig. 3a and Supplementary Fig. 3a). This system allowed us to capture, with high temporal resolution, the transition from NMPs to nascent PSM that would be otherwise difficult to pinpoint in vivo due to the limited number of progenitors. ScRNA-seq analyses at different time-points identified NMPs by the co-expression of *T-Bra*, *Sox2*, *Wnt3a*, *Nkx1-2* and *Cdx2* at 24 h (Fig. 3b, c and Supplementary Data 3). These NMPs expressed *Cdx1*, *Cdh1*, *Fst* and the anterior *Hox* genes (*Hoxa1-b1*), a signature characteristic of E8.5 NMPs[18]. At 36 and 48 h, NMPs expressing posterior *Hox* genes (*Hox6-9*) compatible with a E9.5 NMP or pre-neural signature coexist with cells transitioning towards PSM[18], marked by *Tbx6*, *Foxc2* and *Meox1* (Fig. 3b, c). To confirm that our in vitro model replicated genuine in vivo PSM differentiation, we assessed the capacity of these cells to generate somite-derived tissues, by differentiating PSM towards sclerotome and dermomyotome, which give rise to axial skeleton and muscles respectively (Supplementary Fig. 3b). Populations of sclerotome-like cells expressing *Foxc2*, *Pax1* and *Pax9*, and dermomyotome-like cells marked by *Pax3* and *Pax7* were readily detectable following 5 days of differentiation, confirming the potency of the progenitors (Supplementary Fig. 3c, d).

To address PBX role in PSM formation, we established *Pbx-DKO* cell lines, either by inactivating *Pbx1* and *Pbx2* using CRISPR/Cas9 (Fig. 3d) or by CRE-mediated recombination of *Pbx1*[flox/flox]*;Pbx2*[−/−] derived EpiSCs (Supplementary Fig. 3e). Real-time quantitative PCR (RT-qPCR) and IF revealed impaired ability of *Pbx-DKO* cells to generate dermomyotome and sclerotome, mimicking the *Pbx-DKO* and *Pbx-com* embryonic defects (Supplementary Fig. 3f–h)[24,25].

We used scRNA-seq analyses along PSM differentiation to test if *Pbx-DKO* phenotypes are recapitulated in vitro (Fig. 3e), focussing on the transition from NMPs to PSM. Temporal trajectory reconstruction along differentiation showed that wild-type (WT) NMPs efficiently differentiate towards PSM, whereas *Pbx-DKO* cells fail to reach the PSM state and accumulate as progenitors co-expressing *T-Bra* and *Sox2* (Fig. 3f, g). IF staining and quantification confirmed that PBX loss strongly impairs PSM differentiation, resulting in a higher proportion of NMPs and MPCs at the expense of PSM (Fig. 3h, i), recapitulating the phenotypes observed in vivo and highlighting the critical role of PBX proteins in specifying PSM.

**PBX proteins are dynamically recruited to PSM genes**. To identify PBX targets during PSM differentiation, we performed chromatin immunoprecipitation followed by deep sequencing (ChIP-seq) using PBX1 and PBX2 antibodies at different time-points (Fig. 4a). The number of PBX1 peaks increased during differentiation (Supplementary Fig. 4a), with a maximum number of binding events occurring at 24 h: the stage when mesodermal fate is established. The functional readout of PBX-binding events was assessed by RNA-sequencing (RNA-seq) of WT and *Pbx-DKO* cells at 12 and 24 h (Fig. 4a–c and Supplementary Fig. 4b, c). To pinpoint the genes directly modulated by PBX, we integrated the ChIP-seq and RNA-seq datasets and identified 251 differentially regulated targets (Supplementary Data 4). GO-term analyses of the downregulated genes confirmed PBX requirement in PSM differentiation, somite development and segmentation (Fig. 4b and Supplementary Fig. 4b). By contrast, loss of PBX resulted in increased expression of neuronal and NMP genes like *Sox2* and *Cdx2*, suggesting that PBX may be important for balancing the alternative fates of NMPs (Fig. 4c and Supplementary Fig. 4c). As previous reports have suggested that PBX proteins possess pioneer factor activity[29–31], we assessed the PBX-dependent changes in chromatin accessibility at the PBX-bound sites. We performed Assay for Transposase-Accessible Chromatin using sequencing (ATAC-seq) at different time-points (Fig. 4d) and compared chromatin accessibility of the PBX target genes misregulated in *Pbx-DKO* cells (Fig. 4e, f). Regulatory regions associated with upregulated genes in *Pbx-DKO* were open in EpiSCs and remained accessible in differentiated WT and *Pbx-DKO* cells (Fig. 4f). By contrast, loci associated with downregulated genes in *Pbx-DKO* were closed in EpiSCs and became increasingly accessible during PSM differentiation peaking at 24 h in WT, but not in *Pbx-DKO* cells (Fig. 4e). Thus, by enabling chromatin opening at PSM loci, PBX proteins could prime regulatory regions for the recruitment of lineage-specific TFs. These results substantiate the previously suggested role of PBX as pioneer factors[29–31], but raise the question of how these broadly expressed proteins act at specific loci. As canonical transcriptional regulation is known to be driven by cooperativity, we searched for factors that could cooperate with PBX to promote chromatin accessibility at key PSM regulatory regions[32]. De novo motif analyses of the PBX-bound regions revealed dynamic changes in the TF-binding site enrichment during differentiation, suggesting cooperative binding with distinct partners at different time-points (Fig. 4g and Supplementary

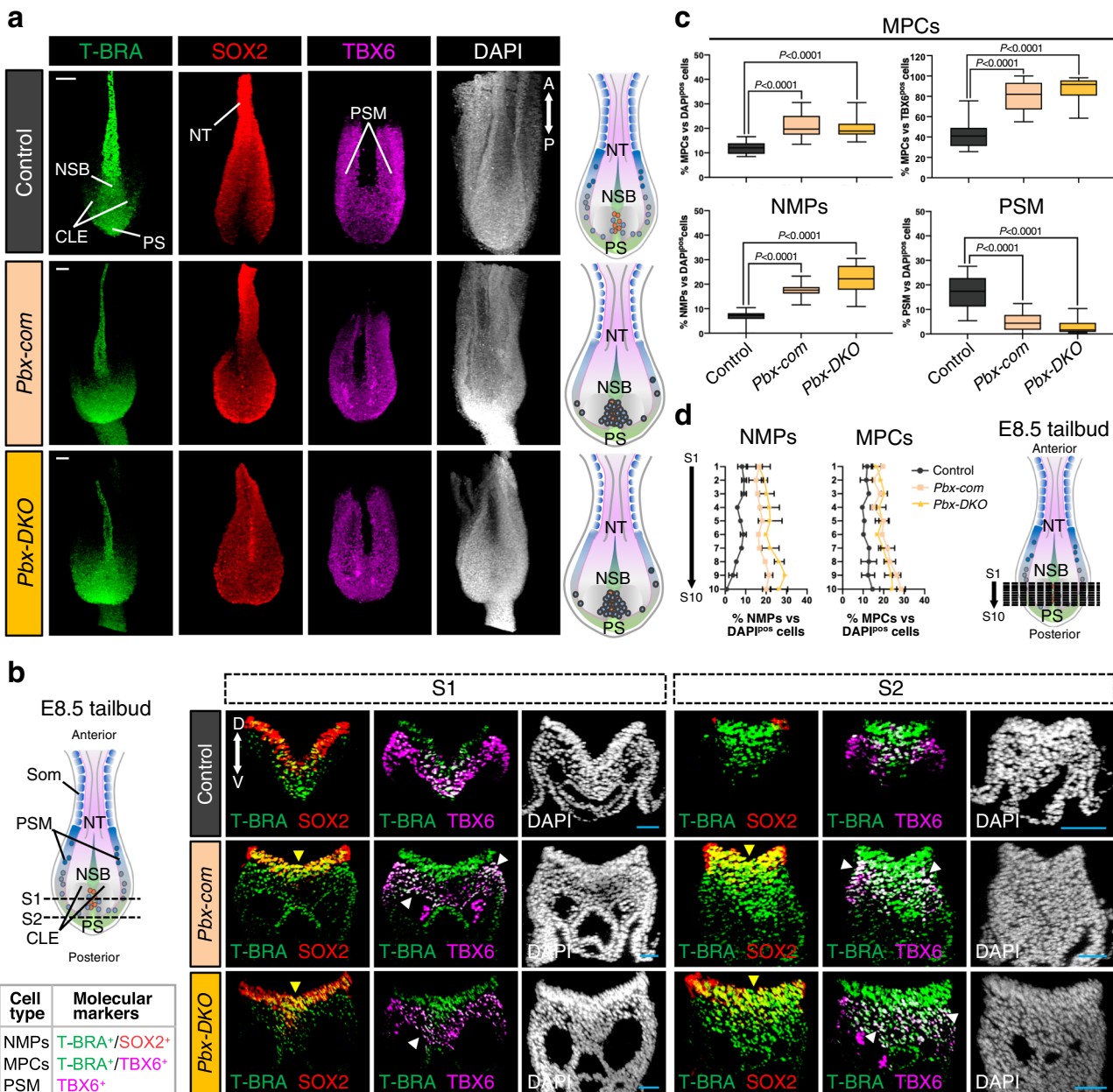

**Fig. 2 *Pbx* mutant NMPs fail to acquire paraxial mesodermal fate, do not progress towards PSM, and accumulate in the tailbuds. a** Confocal maximum intensity projection images of E8.5 tailbuds probed with T-BRA (green), SOX2 (red) and TBX6 (magenta) antibodies and counterstained with DAPI (grey) reveal the enlarged CLE and accumulation of T-BRA^pos, SOX2^pos, TBX6^pos cells in *Pbx-com* and *Pbx-DKO* tailbuds. The cartoons on the right depict dorsal views of E8.5 control and *Pbx* mutant tailbuds, with NMPs represented as red dots and MPCs as blue dots. Scale bars: 50 μm. **b** Confocal transverse sections of control, *Pbx-com* and *Pbx-DKO* tailbuds through the caudal progenitor zone. Black dashed lines in the cartoon indicate the plane of sections. Representative sections S1 and S2 reveal accumulation of T-BRA/SOX2 double-positive (T-BRA^pos;SOX2^pos) NMPs in *Pbx* mutants (yellow arrowheads). TBX6 staining shows reduced expression in PSM and co-localisation with NMP markers in *Pbx-com* and *Pbx-DKO* tailbuds (white arrowheads). The panel on the left summarises the molecular signature used to identify the different cell types along the path towards PSM. Scale bars: 30 μm. **c** Box plots assessing the proportion of T-BRA^pos;TBX6^pos cells (MPCs) relative to DAPI^pos cells (total cells) or to TBX6^pos cells (PSM), and T-BRA^pos;SOX2^pos cells (NMPs) and TBX6^pos cells (PSM) relative to DAPI^pos cells. The box contains the 25th to 75th percentiles of the dataset, the centre line denotes the median value (50th percentile) and the whiskers extend from the smallest to the largest value. *P*-values were calculated by one-way ANOVA and Tukey's test for multiple comparisons and are indicated in the figure. **d** Distribution of NMPs and MPCs along the anterior/posterior axis shows accumulation of NMPs and MPCs in the posterior tailbud of *Pbx-com* and *Pbx-DKO* embryos. Data are mean ± s.e.m. Black dashed lines in the cartoon indicate the plane of sections. In **c** and **d**, the percentage of different cell types for each embryo was calculated from 10 independent sections equally distributed along the anterior/posterior axis. In **a**, **b** and **d**: CLE caudal lateral epiblast, NSB node-streak border, PS primitive streak, PSM pre-somitic mesoderm, Som somites, NT neural tube, A anterior, P posterior, D dorsal, V ventral.

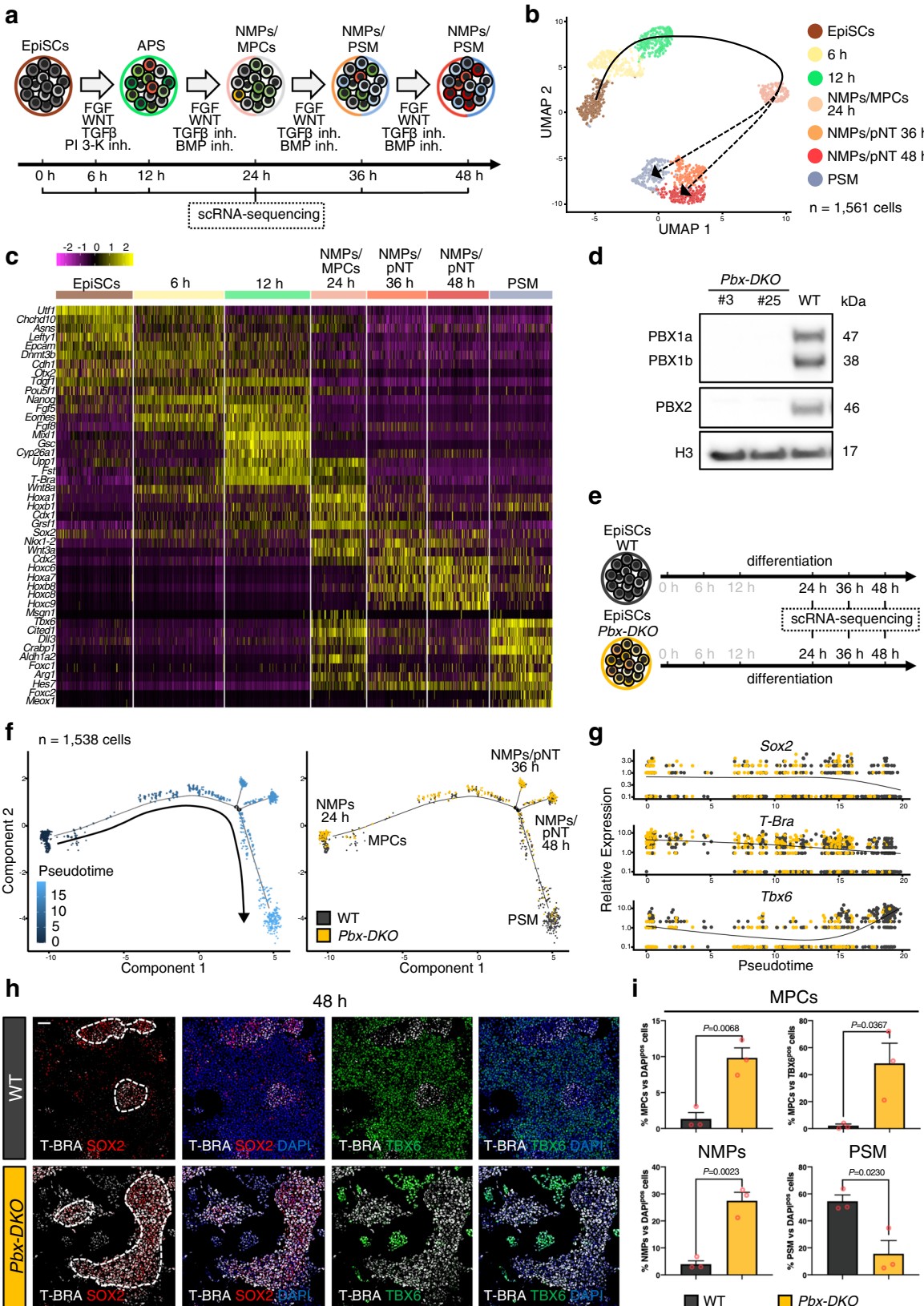

Data 5). In EpiSCs and at 6 h, when HOX are absent, we identified the PBX3 and PREP1 binding motifs[33], whereas at later time-points, once *Hox* genes are expressed, the HOX recognition sites featured prominently. Notably during the 12-24 h time window that includes the NMP-to-PSM transition, we identified LEF1 among the top consensus motifs nearby the PBX-bound regions. These data suggest that LEF1 and PBX might bind cooperatively to PSM genes, and provided a possible reason why *Pbx-DKO* cells cannot complete PSM maturation despite the presence of WNT ligands (Supplementary Fig. 2d).

**Fig. 3** *Pbx* **mutant cells fail to acquire paraxial mesoderm identity in vitro. a** Schematics of the in vitro protocol used to differentiate EpiSCs towards anterior primitive streak (APS) and pre-somitic mesoderm (PSM). **b** ScRNA-seq of in vitro PSM differentiation. Cells were harvested at different time-points, as indicated in **a**. Unsupervised clustering and UMAP visualisation reveal temporally distinct populations of NMPs (orange-red), MPCs (light pink) and PSM (light blue). Individual cells are coloured according to annotated cluster identities. Reconstructed developmental trajectories are indicated with black dashed lines. **c** Heatmap showing expression of key genes along the path towards PSM. Cells acquire a NMP signature at 24 h, and subsequently undergo progressive maturation towards PSM. NMPs present at 36–48 h express higher levels of posterior *Hox* genes (*Hox6-9*), consistent with the acquisition of more posterior axial identity. Colour bar indicates the intensity associated with normalised expression values. **d** Western blot analysis confirms absence of PBX1 and PBX2 proteins in the *Pbx-DKO* cell lines obtained by inactivating *Pbx1* and *Pbx2* in 129 EpiSCs with a CRISPR/Cas9 approach. **e** Schematics of wild-type (WT) and *Pbx-DKO* EpiSCs differentiation towards PSM. The time-points selected for scRNA-seq analyses are highlighted in black. **f** Monocle's pseudotemporal ordering of WT (black) and *Pbx-DKO* cells (yellow) spanning the NMP-to-PSM transition (24–48 h), showing inability to generate PSM and increased number of NMPs/MPCs caused by loss of PBX. **g** Pseudotime analysis of PSM differentiation showing distribution of WT (black) and *Pbx-DKO* cells (yellow) expressing the NMP genes *Sox2* and *T-Bra* and the PSM marker *Tbx6*. **h** Representative immunofluorescence staining for T-BRA (white), SOX2 (red) and TBX6 (green) at 48 h of differentiation reveals an accumulation of NMPs (T-BRA$^{pos}$;SOX2$^{pos}$, dashed white lines) in *Pbx-DKO* cells. Scale bar: 50 μm. **i** Histograms representing the percentage of progenitors in WT (black) and *Pbx-DKO* cells (yellow) at 48 h of differentiation. Upper panel: percentage of MPCs (T-BRA$^{pos}$;TBX6$^{pos}$) relative to DAPI$^{pos}$ cells (total cells) or to TBX6$^{pos}$ cells (PSM). Lower panel: percentage of NMPs (T-BRA$^{pos}$;SOX2$^{pos}$) and PSM (TBX6$^{pos}$) cells relative to DAPI$^{pos}$ cells. Data are mean ± s.e.m. *P*-values indicated in the figure were calculated by two-tailed unpaired *t*-test (*n* = 3 biological replicates).

**PBX proteins are interpreters of LEF1-mediated WNT signalling.** Like PBX, the WNT effectors (TCFs/LEF1) exhibit low binding specificity[34]. Therefore, the observation that PBX and LEF1 could be part of a common regulatory module supported the hypothesis of a PBX-LEF1 DNA-binding cooperativity at PSM genes. LEF1 and TCF7 are the most prominent WNT effectors involved in paraxial mesoderm development, but while the latter is broadly expressed in the tailbud, LEF1 is specifically localised in PSM and posterior somites[35] (Fig. 5a). Notably, LEF1 expression is expanded in E8.5 *Pbx-DKO* tailbuds, suggesting that the *Pbx* mutant phenotypes are not due to a reduction in LEF1 levels (Fig. 5a). To assess whether PBX and LEF1 TFs cooperatively control common transcriptional targets, we exploited our in vitro differentiation system, and performed ChIP-seq for LEF1. We found that LEF1 is de novo and progressively recruited to chromatin between 12 and 24 h, concomitantly with the formation of MPCs (Fig. 5b). Remarkably, LEF1 recruitment is abolished in mutant cells (Fig. 5b), despite sustained LEF1 expression (Fig. 5a and Supplementary Fig. 5a). By integrating our ChIP-seq datasets for PBX and LEF1 (Fig. 5c and Supplementary Data 6) and RNA-seq of WT and *Pbx-DKO* cells at 24 h (Supplementary Fig 5b and Supplementary Data 6), we observed that over 60% of the LEF1 targets (including *Sox2*, *T-Bra*, *Cdx2*, *Msgn1* and *Aldh1a2*) were co-bound by PBX (Fig. 5c) and misregulated in *Pbx-DKO* (Supplementary Fig. 5b). As expected, GO-term analyses underscored targets associated with WNT activity and somitic features (Fig. 5d). We observed that while the occupancy of PBX and LEF1 occurs at variable distances on different regulatory regions, the majority (61%) of the LEF1 ChIP-seq peaks falls in close proximity to the PBX peaks (Supplementary Fig. 5c). ATAC-seq analyses of PBX-LEF1 common targets revealed that chromatin accessibility at LEF1-bound regions decreased for downregulated genes (Supplementary Fig. 5d) and remained unchanged for upregulated genes (Supplementary Fig. 5e), as illustrated for *Msgn1*, *Aldh1a2*, *Sox2* and *Cdx2* (Fig. 5e). These data demonstrate that PBX proteins authorise WNT-dependent transcription by making PSM genes accessible to LEF1. Instead, in the absence of PBX, the inability to activate the PSM fate programme triggers the feedforward mechanisms sustaining NMP expansion.

**Distinct PBX complexes are sequentially recruited to the *Msgn1* regulatory region.** To establish how the PBX-LEF1 module operates, we focussed on the master regulator of the paraxial mesoderm fate, *Msgn1*, which controls both cell migration and differentiation[10]. Undifferentiated progenitors accumulate in *Msgn1* mutant tailbuds comparably to *Pbx-DKO* embryos[10,14]. Notably, *Msgn1* expression was lost in *Pbx-com* embryos (Fig. 6a) and strongly downregulated in *Pbx-DKO* cells upon PSM differentiation (Fig. 6b). Moreover, the MSGN1 target *Snai1*[10] was downregulated in E8.5 *Pbx-DKO* and *Pbx-com* PSM (Supplementary Fig. 6a), suggesting that epithelial-to-mesenchymal transition (EMT) and cell migration are affected in absence of PBX. To assess the cell migratory capacity of WT and mutant MPCs, we visualised PSM differentiation in time-lapse videos (Supplementary Movies 1 and 2 and Supplementary Fig. 6b). We observed that WT cells leave the NMP zone upon differentiation, while *Pbx-DKO* cells fail to migrate and accumulate in the expanding progenitor area. Lentiviral re-expression of *Msgn1* in *Pbx-DKO* cells during differentiation resulted in downregulation of NMP genes and upregulation of PSM genes (Supplementary Fig. 6c). These data confirmed that PBX-LEF1 direct PSM development and EMT by controlling *Msgn1* expression.

*Msgn1* is the earliest expressed paraxial mesoderm gene and is tightly regulated in vivo and in vitro, being transcribed in the 12–36 h window corresponding to the NMP-to-PSM transition (Fig. 6b). ChIP-seq data showed that PBX and LEF1 are dynamically recruited to the *Msgn1* promoter, mirroring *Msgn1* temporal expression, and LEF1 occupancy is lost in *Pbx-DKO* cells (Fig. 6c).

To identify the factors that temporally recruit LEF1 on *Msgn1* promoter, we focussed on PBX and their combinatorial logic[36]. Based on the temporal expression of HOX and TALE family members, we identified several candidates such as the hetero-dimerizing partner *Meis2* and group-1 *Hox* genes (*Hox-1*) (Supplementary Fig. 7a) that are expressed during the period of PBX occupancy at the *Msgn1* promoter. In addition, a second PBX-interacting factor, PREP1, is expressed earlier than *Msgn1* (Fig. 6d). Considering the MPC-specific enrichment of *Pbx1*, *Meis2* and *Hox-1* (Fig. 6e) these proteins represent the best candidates for regulating *Msgn1*. We hypothesised that two putative complexes would be sequentially recruited, with PREP1/PBX binding prior to the PBX/MEIS2/HOX-1 complex. Consistent with this idea, ChIP-qPCR showed that PREP1 is recruited to *Msgn1* first, at 12 h, while MEIS2 and PBX are recruited later, at 12–24 h (Fig. 6f). Examining the *Msgn1* regulatory region, we identified a PBX/HOX-1 responsive site[37] within the region of PBX occupancy (Fig. 6g and Supplementary Fig. 7b). Electrophoretic Mobility Shift Assay (EMSA) with either in vitro-

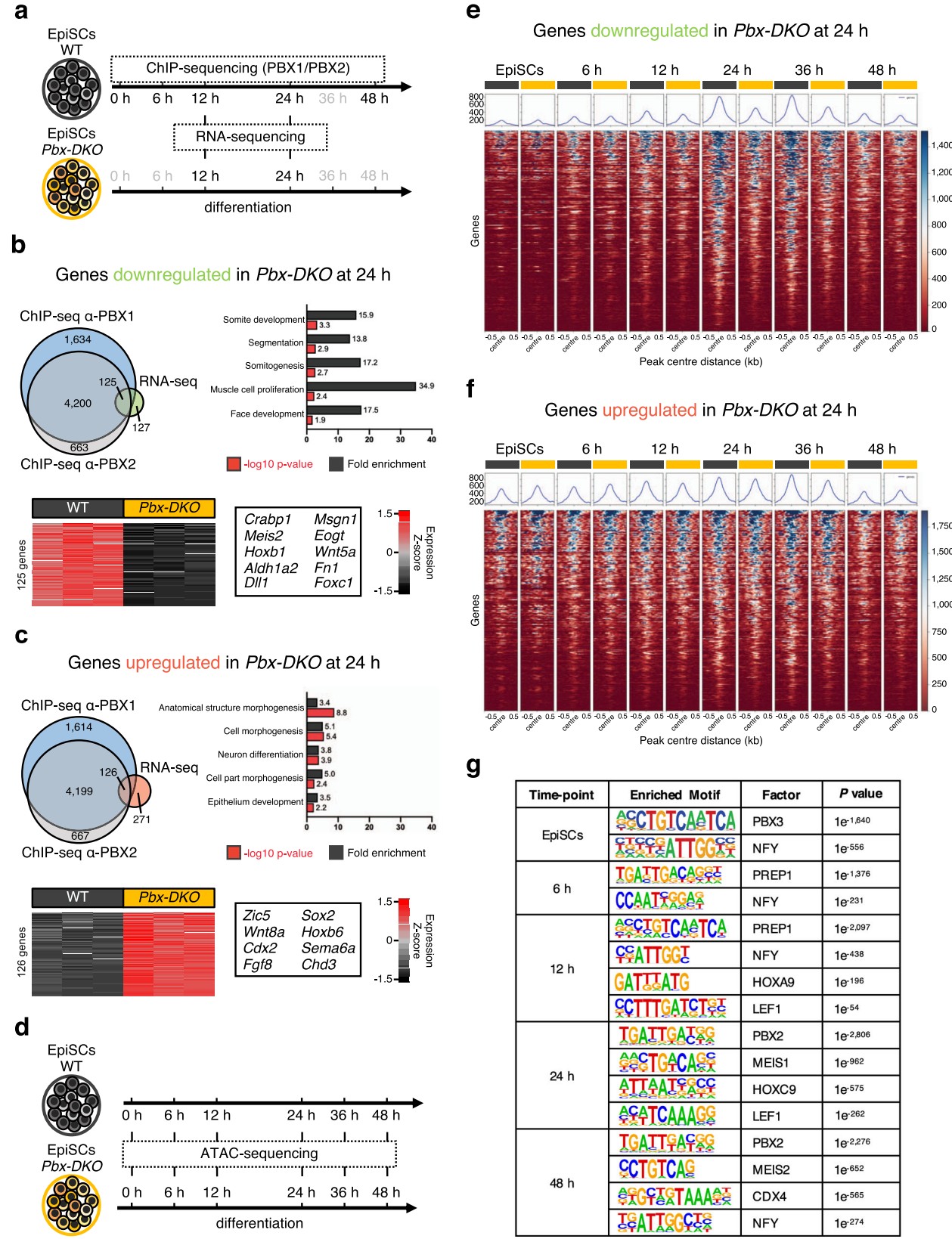

translated proteins or 24 h nuclear extracts highlighted the PBX/MEIS2/HOX-1 ternary complex formation on the oligonucleotide containing the PBX/HOX-1-binding site (*Msgn1-P2*) but not on its mutated variant (*Msgn1-P2-M*) (Fig. 6h). In agreement with the sequential recruitment to the *Msgn1* regulatory region, we found that while the assembly of PBX/MEIS2 on *Msgn1-P2*

depended on the presence of HOX-1 (Supplementary Fig. 7c), the PREP1/PBX heterodimers could bind in its absence (Supplementary Fig. 7d). These binding properties suggest that the expression of HOX-1 drives the assembly of a new complex at the *Msgn1* promoter that would be responsible for LEF1 occupancy and WNT-mediated activation. Consistent with this idea, affinity

**Fig. 4 PBX proteins occupy regulatory regions associated with PSM genes and remodel chromatin accessibility. a** Schematics of EpiSC differentiation towards PSM. The time-points selected for ChIP-seq/RNA-seq analyses are highlighted in black. **b** Top: overlap of downregulated genes in *Pbx-DKO* cells (RNA-seq) and genes associated with PBX-occupied regions at 24 h (ChIP-seq). GO-term enrichment analysis displays significant overrepresentation of terms associated with mesoderm and somite development (Binomial test with Bonferroni correction). Bottom: heatmap showing RNA-seq expression Z-scores computed for 125 downregulated PBX targets in *Pbx-DKO* (yellow) compared to WT cells (black). Representative markers are indicated. **c** Top: overlap of upregulated genes in *Pbx-DKO* cells (RNA-seq) associated with PBX-occupied regions (ChIP-seq) at 24 h. GO-term enrichment analysis shows correlation with terms associated with neurectoderm and embryonic development (Binomial test with Bonferroni correction). Bottom: heatmap displaying RNA-seq expression Z-scores computed for 126 upregulated PBX targets in *Pbx-DKO* (yellow) compared to WT cells (black). Representative markers are indicated. In **b** and **c**, significance was assessed by DEseq2 on the basis of two-sided Wald test with Benjamini–Hochberg adjusted *P*-values ($P \leq 0.05$, fold-change $\geq 1.5$). $n = 3$ biological replicates. **d** Schematics of EpiSC differentiation towards PSM illustrating the time-points selected for chromatin accessibility analysis (ATAC-seq). **e** Heatmap of chromatin accessibility of WT (black) and *Pbx-DKO* cells (yellow) at PBX-occupied regions that are associated with downregulated genes in *Pbx-DKO* along PSM differentiation, showing reduced accessibility in *Pbx-DKO* cells at 24 and 36 h. **f** Heatmap of chromatin accessibility at PBX-bound regions associated with upregulated genes in *Pbx-DKO* compared to WT cells. Chromatin accessibility is comparable between WT (black) and *Pbx-DKO* cells (yellow). The scale represents normalised counts (RPKM) for ATAC-seq peak signals ±500 bp around the centre of the peak in **e** and **f**. **g** Homer de novo motif analysis of 200 bp summit regions along differentiation reveals changing in binding sites enrichment at different time-points, highlighting the variety of PBX-binding partners during PSM differentiation. Only the most relevant binding motifs are shown for each time-point. See Supplementary Data 5 for the complete list.

binding assays confirmed that the PBX/MEIS2/HOX-1 complex has higher affinity for *Msgn1-P2* than PBX/PREP1 complexes, suggesting that it could compete for binding once HOX-1 is expressed (Supplementary Fig. 7e–g). Moreover, we observed that changes in the accessibility of *Msgn1* regulatory regions were correlated with *Hox-1* expression, and concurrent with PBX-LEF1 recruitment. Accordingly, in *Pbx-DKO* the accessibility of the loci containing the PBX/HOX and LEF1-binding sites was reduced at 24 h (Fig. 6i), highlighting a potential role of PBX/HOX complexes to nucleate their own binding and their ability to destabilise the nucleosomes. These data uncover a hierarchical TF-binding network on the *Msgn1* locus, but do not conclusively demonstrate that PBX/HOX are directly involved in chromatin remodelling, prompting us to assess the recruitment of additional pioneer factors.

**TALE complexes direct chromatin remodelling at the *Msgn1* regulatory region, synergising with T-BRA.** T-BRA is a pioneer factor essential for priming mesodermal genes[38]. ChIP-qPCR analyses showed that T-BRA is already recruited at 12 h to the LEF1-binding region of *Msgn1*, and its occupancy decreases over time (Fig. 7a). Interestingly, T-BRA occupancy is not abrogated in *Pbx-DKO* (Fig. 7a), suggesting that the binding of T-BRA on *Msgn1* is not sufficient to drive paraxial mesoderm differentiation and that PBX complexes remodel the chromatin independently of T-BRA.

To establish the direct consequence of the binding of PBX complexes on chromatin accessibility and their effects on LEF1 recruitment, we employed CRISPR/Cas9 technology to generate lines carrying the *pMsgn1* point mutations (*pMsgn1-mut*) studied by EMSA (Fig. 6g, h and Supplementary Fig. 8a, b), preventing the binding of PBX complexes to the *Msgn1* promoter but allowing their recruitment to other targets. RT-qPCR analyses showed that the expression of *Msgn1* and its downstream targets were affected in *pMsgn1-mut* cells (Fig. 7b), and PSM differentiation was impaired (Fig. 7c), thus recapitulating at least in part the *Pbx-DKO* phenotype. However, the mesodermal phenotype of *pMsgn1-mut* lines was exacerbated, as shown by the absence of *T-Bra*, the paraxial mesodermal gene *Tbx6*, and the EMT markers *Cdh2* and *Snai1*. ChIP-qPCR revealed that in *pMsgn1-mut* neither PBX nor LEF1 were recruited to *Msgn1* (Fig. 7d). Interestingly, the recruitment of T-BRA was also affected (Fig. 7d) in *pMsgn1-mut*, and ATAC-seq analyses showed that both LEF1 and PBX/HOX-binding regions were inaccessible in *pMsgn1-mut* cells (Fig. 7e), unequivocally confirming the role of PBX complexes as pioneer factors and modulators of chromatin accessibility. Furthermore, the reduced

recruitment of T-BRA reveals that target gene selection specificity is achieved through the collaborative effort of multiple pioneer TFs.

In conclusion, paraxial mesoderm induction relies on a WNT-HOX integrated code that is coordinated by the TALE complexes, the WNT-effector LEF1 and the synergistic activity of T-BRA. While T-BRA broadly assesses mesodermal genes, TALE/HOX-1 interpret WNT signalling response by specifically recruiting LEF1 at the PSM genes, resulting in paraxial mesoderm differentiation via a *Msgn1*-mediated transcriptional programme.

## Discussion

In this work, we dissected how TALE complexes control the GRNs that alter the NMP response to WNT signalling and activate the paraxial mesoderm programme. We established that TALE/HOX and WNT signalling work through a common genetic code that simultaneously represses NMP genes and activates the PSM differentiation programme. We found that switching of the PBX-binding partners result in TALE/HOX high-affinity complex formation that destabilises the nucleosomes at WNT-responsive elements enabling de novo recruitment of the WNT-effector LEF1, thereby challenging the consolidated model of WNT-mediated response centred on the pre-bound activity of TCFs/LEF.

This concerted TALE-LEF1 activity represses NMP genes (like *Sox2*, *T-Bra* and *Cdx2*), ensuring silencing of the neural and NMP transcriptional programmes in PSM-fated cells. Concurrently, TALE-LEF1 synergistically drive the expression of PSM genes including the master regulator *Msgn1*, which in turn controls EMT and the migration of MPCs from the progenitor zone to the anterior PSM. Loss of the TALE activity results in the accumulation of progenitors trapped in a NMP/MPC state that ultimately causes an enlarged tailbud and impaired somitogenesis (Fig. 8a).

In vertebrates, *Hox* genes are collinearly expressed during axial elongation, establishing the HOX code with a spatio-temporal gradient along the anterior-posterior axis[19,20,39]. This sequential activation provides NMPs with positional information and prompts MPCs to progressively adopt a more posterior identity[11], ensuring that cell fate and position are intrinsically linked. Our data suggest that group-1 HOX are also indispensable for the initial specification of MPCs and their migration to form PSM; thereby balancing the residence time in the progenitor zone and the flux of PSM patterning the trunk region. In *Pbx* mutants, 5′ *Hox* genes are overexpressed (Supplementary Fig. 1h) in NMP/MPCs, suggesting the existence of progenitors with a posterior identity and thereby a potential posteriorisation of the PSM, in

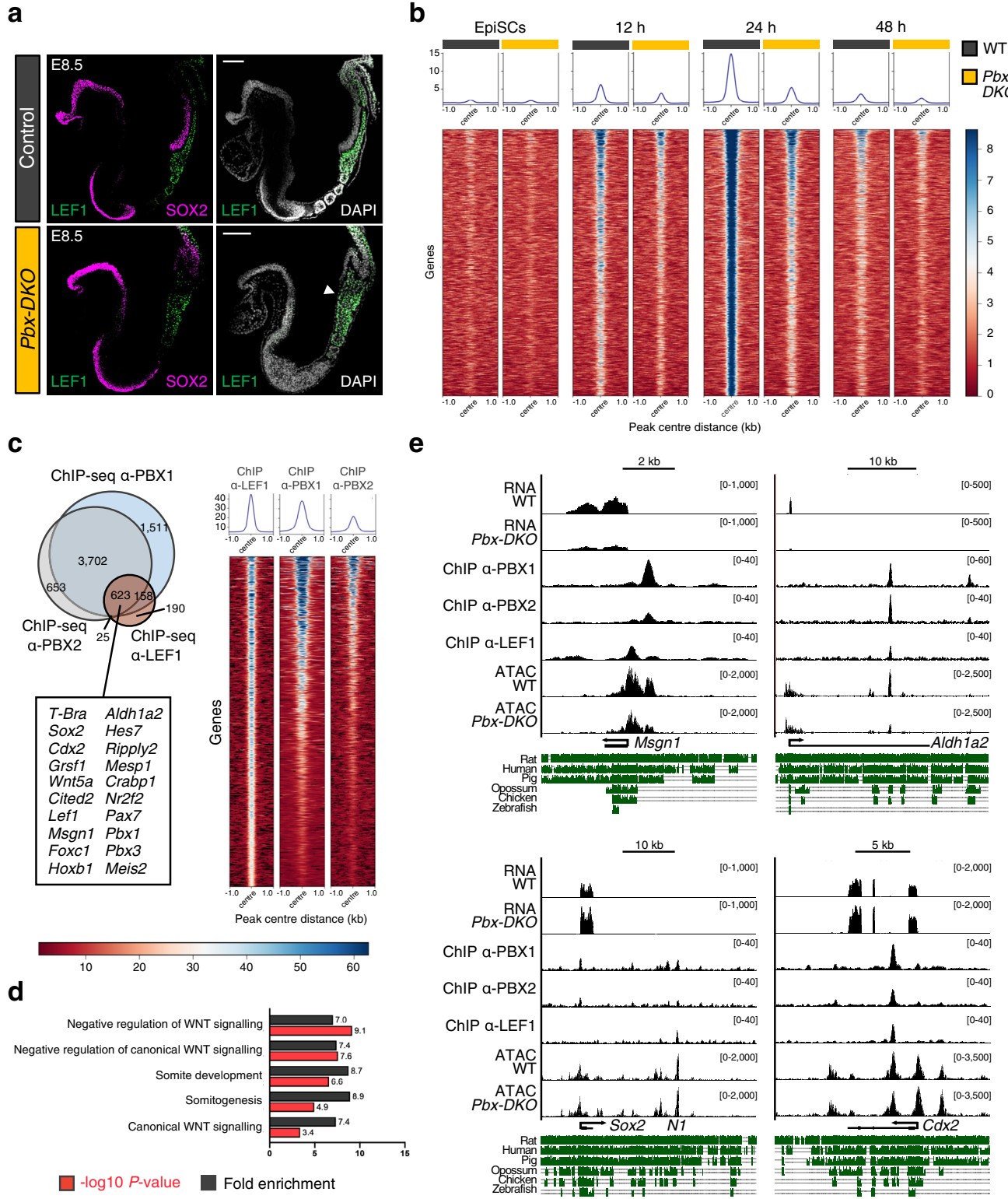

agreement with the observation of a rostral shift of the axial *Hox* gene expression in the *Pbx-com* mutants[25].

In addition to providing HOX with binding specificity[37], PBX have also been indicated as pioneer factors[29,31]. However, a key unanswered question is how such broadly expressed TFs can exquisitely open lineage-specific loci in a space and time-dependent manner. Exploiting the cooperative interactions among TALE and HOX proteins, and focussing on the tightly regulated spatio-temporal expression of *Msgn1*, we uncovered the

sequential recruitment of TALE factors generating, at least in vitro, a high-affinity (PBX/MEIS2/HOX-1) ternary complex that could promote chromatin accessibility of the *Msgn1* promoter (Fig. 8b). While the binding of the PBX/MEIS2 complex occurs concomitantly with increased chromatin accessibility and LEF1 recruitment, the low-affinity PBX/PREP1 dimers are recruited earlier, suggesting that they could be important to scan the DNA or increase the local concentration of PBX proteins, generating the TALE hubs that assist the loading of the TALE/

**Fig. 5 PBX complexes remodel the chromatin environment for LEF1 recruitment. a** Representative immunofluorescence staining for LEF1 (green) and SOX2 (magenta) on sagittal sections of E8.5 mouse embryos, revealing specific localisation of LEF1 in PSM and posterior somites of control and *Pbx-DKO* embryos. LEF1 expression is bloated in the tailbud of *Pbx-DKO* (white arrowhead). DAPI was counterstained in grey. Scale bars: 100 μm. **b** Heatmap of LEF1 ChIP-seq signals of WT (black) and *Pbx-DKO* cells (yellow) reveals PBX crucial role in LEF1 recruitment, as demonstrated by the extensive drop of LEF1 occupancy in *Pbx-DKO* cells along differentiation. **c** Left: overlap of genes bound by PBX1 (light blue), PBX2 (grey) and LEF1 (brown). A representative subset of target genes co-bound by PBX and LEF1 is reported in the lower box. Right: heatmap of tag densities of LEF1, PBX1 and PBX2 ChIP-seq peaks at the co-bound regions identified at 24 h of differentiation. In **b** and **c**, the scale represents normalised counts (RPKM) for ChIP-seq peak signals ±1 kb around the centre of the peak. **d** GO-term enrichment analysis of genes bound by PBX and LEF1 displays significant overrepresentation of terms associated with WNT signalling and somite development (Binomial test with Bonferroni correction). **e** RNA-seq, ChIP-seq and ATAC-seq coverage tracks for PBX-LEF1-activated (*Msgn1*, *Aldh1a2*) and repressed (*Sox2*, *Cdx2*) target genes. Threshold of vertical viewing range of data based on RPKM values is noted. Conservation across vertebrates is indicated in green.

HOX complex[40]. The observation that PBX are pre-bound on regulatory regions prior to gene activation[29–31] has been used as proof for their pioneering activity. However, there are no data conclusively showing that they unlock closed chromatin. The pioneer activity of TFs is usually determined by measuring reduced chromatin accessibility in wild-type and knockout cells[38], that however can also be caused via indirect mechanisms. Here, using precise CRISPR-mediated mutagenesis, we unequivocally demonstrate that point mutations affecting the binding of PBX/HOX to the endogenous *Msgn1* promoter reduce chromatin accessibility and LEF1 recruitment, indicating that PBX directly modulate chromatin accessibility and context-dependent WNT responses.

Analyses of PBX and LEF1 occupancy revealed that the distance between them is flexible and varies at different regulatory regions. This arrangement could support cooperative binding, allowing the specific recruitment of TFs that further stabilise the nucleosome-depleted regions triggered by the PBX complexes. Thus, changes in chromatin accessibility initiated by the PBX complexes could mediate recruitment of additional factors, including ATP-dependent chromatin remodelling factors. PBX/HOX could interact with histone modifying complexes like CBP, TrxG or HDAC and PcG proteins[41] or may affect chromatin conformation via recruitment of histone variants[42]. TF-binding sites located in closed chromatin could provide an additional layer of specificity by restricting the access to highly specific PBX/HOX complexes[43,44]. Interestingly, the NFY-binding site appeared among the most enriched motifs identified within the PBX-bound regions at all time points of differentiation[45]. In zebrafish, NFY has been demonstrated to directly interact with the TALE proteins to promote access to enhancers, suggesting that it could play a similar role in paraxial mesoderm differentiation. For all these reasons, it is plausible that chromatin landscape changes are driven by combined effects on DNA-histone affinity and recruitment of nucleosome remodelling factors.

Intriguingly, the TALE-WNT integrated code is conserved in the regulatory regions of PSM genes within eutherians where axial elongation relies on expanding NMPs, but is absent in lower vertebrates where PSM maturation is supported by depleting progenitors[6,46]. Of note, the TALE-WNT integrated motif is present at the *Sox2* enhancer *N1*[47] that is exclusively associated with expansion of NMPs in amniotes. Thus, during vertebrate evolution, WNT, TALE and HOX proteins could have been co-opted by eutherians to balance expansion and differentiation of paraxial mesoderm progenitors for axial elongation.

Prevailing models of WNT activity suggest that TCFs/LEF work through a transcriptional switch, binding and repressing genes in unstimulated conditions and promoting transcription upon WNT induction[34]. Here we observed that during paraxial mesoderm induction, LEF1 is de novo recruited when WNT signalling is induced and when the PSM genes become accessible.

Thus, in this context, the competence for WNT signalling response is established during the transition of NMPs to PSM. This alternative mechanism supports the concept that distinct modes exist to prime chromatin accessibility and lineage choice. Crucially, we found that signalling response is dictated by TFs unlocking the WNT-responsive elements rather than by WNT ligand dosage, questioning the current concentration-dependent morphogen model[48].

Given that WNT/LEF1 signalling is important for homeostasis and regeneration of the skin[49], it will be compelling to address if the TALE/HOX motif also unlocks the regenerative transcriptional programmes in the skin, thus having potential therapeutic applications. Furthermore, abnormal activity within this pathway is linked to hyperproliferation and EMT-driven metastatic processes[50]. In this context, it will be important to establish if aberrations in TALE-WNT integrated codes could also be associated with carcinogenesis and cancer invasion[51].

Finally, the observation that TALE complexes collaborate with the pioneer factor T-BRA[38] provides a temporal code for WNT-mediated response. While T-BRA has a pan-mesodermal role, the TALE complexes provide specificity towards the paraxial mesodermal fate. Hence, conventional cooperativity can be extended to pioneer factor clusters that synergistically generate a permissive chromatin landscape and collectively operate as key determinants of lineage specification. Given that the GRNs governing paraxial mesoderm differentiation[17] are conserved between human and mouse, these findings can be used to improve the current protocol of hESCs differentiation.

Altogether, this work uncovers a cooperative TALE/HOX-WNT integrated code unlocking the spatio-temporal GRNs that control axial extension. During axial elongation, similar modules could also be employed by 5' HOX proteins to further fine-tune WNT signalling response to provide MPCs with a more posterior fate. Thus, integrating *Hox* collinearity and WNT signalling on a common genomic element may represent a powerful strategy that confers mesodermal progenitors with a specific spatio-temporal identity.

## Methods

**Cell Lines**. EpiSC lines were maintained in six-well cell-culture plates (Corning, 3506) pre-coated with 15 μg/ml human Fibronectin (Merck Millipore, FC01015) in N2B27 medium (prepared as described in Ying et al.[52], supplemented with 20 ng/ml Activin A, 10 ng/ml bFGF and 1 μM XAV939 (EpiSC medium). All lines used in this study were routinely karyotyped and tested for mycoplasma.

**Mice**. The mutant alleles used in this study were previously described: *Pbx1* conditional allele in Koss et al.[53] and *Pbx2*−/− mice in Selleri et al.[54] The *Cre* deleter strain, *Sox2^Cre* knock-in mouse[55], was utilised for *Pbx1* inactivation in the inner cell mass. All animals used in this study were maintained in laboratory animal housing facilities at macroenvironmental temperature and humidity ranges of 20–24 °C and 45–65%, respectively, with a 12/12 h light/dark cycle. Embryos were dissected at E8.5/E9.0 and genotyped by PCR as described in Koss et al.[53] All animal work was carried out in accordance with European legislation. All work was authorised by and carried out under the

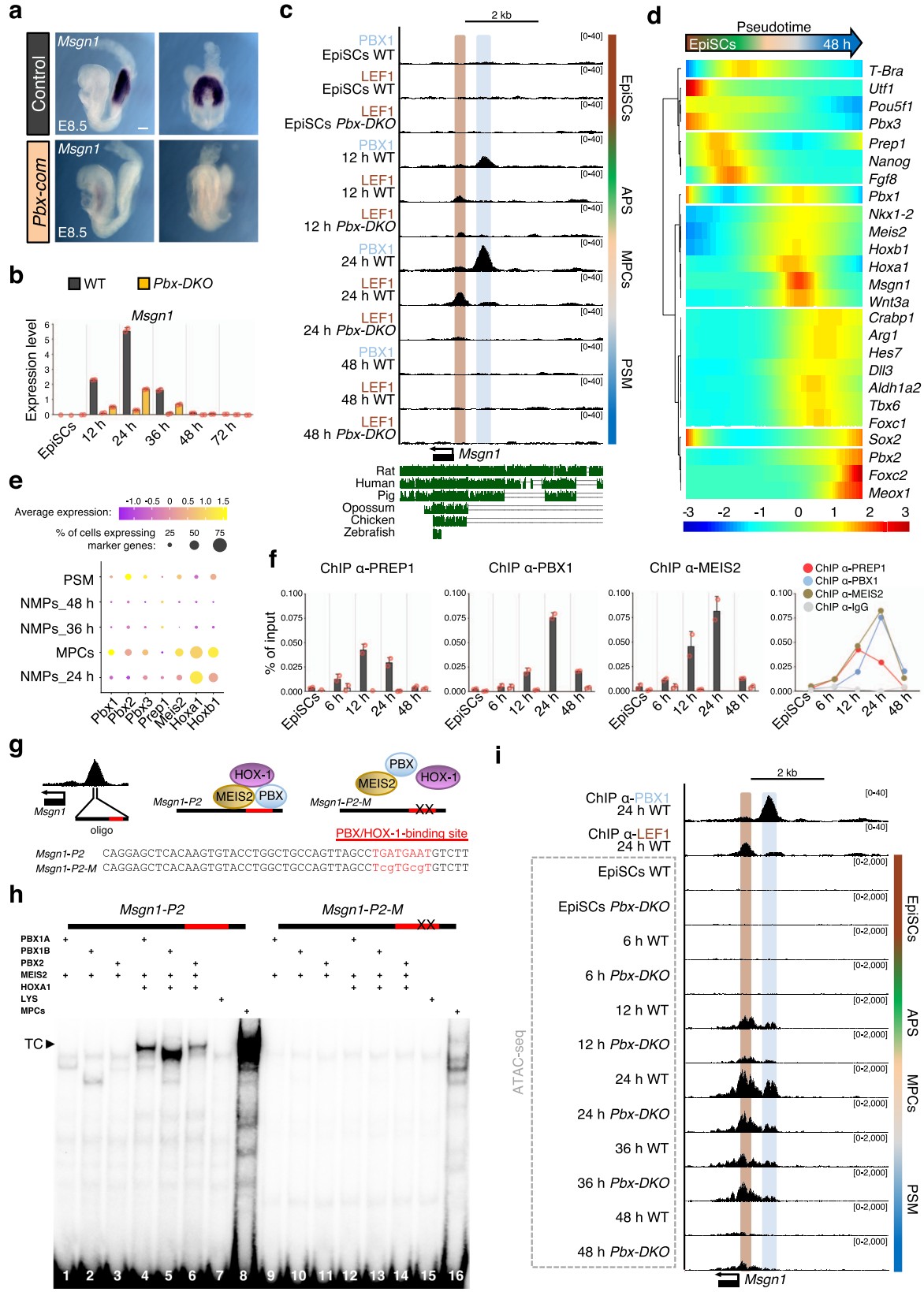

Project License 2017-15-0201-01255 issued by the Danish Regulatory Authority.

**Generation of *Pbx-DKO* lines using the CRISPR/Cas9 system**. EpiSC lines derived from E6.5 mouse embryos (129S6/SvEvTac background) were a kind gift from Dr. M.M. Rothová (The Novo Nordisk Foundation Center for Stem Cell Biology,

University of Copenhagen, Copenhagen, Denmark). The lines were derived in EpiSC medium as previously described[56]. *Pbx-DKO* lines were established via sequential CRISPR/Cas9-mediated gene targeting. First, *Pbx2-KO* lines were generated by inserting a floxed CMV-hygromycin-Poly (A) cassette into Exon 1 of *Pbx2* to disrupt *Pbx2* gene expression, employing the CRISPR Nickase system[57]. Guide RNAs (gRNAs) were designed using the CRISPR/Cas9 design tool (crispr.mit.edu) and cloned into pX335 (a gift from Prof. F. Zhang; Addgene, 42335). Cells were co-transfected with a pair of

**Fig. 6 Sequential recruitment of PBX/HOX-1 complexes to *Msgn1* promoter changes the chromatin landscape and activates transcription. a** In situ hybridisation shows loss of *Msgn1* expression in E8.5 *Pbx-com* tailbuds. Scale bar: 100 μm. **b** Relative *Msgn1* mRNA expression measured by RT-qPCR, revealing dynamic expression during PSM differentiation and strong downregulation in *Pbx-DKO* cells (yellow). **c** UCSC genome browser snapshot of the murine *Msgn1* locus showing conservation across vertebrates. PBX1 (light blue) and LEF1 (brown) are recruited to *Msgn1* promoter at 12–24 h. LEF1 binding is lost in *Pbx-DKO* at any time-points. **d** Pseudotime kinetics of lineage markers along PSM differentiation. *Prep1* is expressed before PS induction (marked by *T-Bra*), while *Pbx1*, *Meis2*, *Hox-1* are transcribed within the same time-window of *Msgn1*. **e** Dot plot analysis of scRNA-seq data of paraxial mesoderm-related clusters showing enriched expression of *Pbx*, *Meis2*, *Hox-1* in MPCs at 24 h. **f** ChIP-qPCR analyses of TALE proteins at indicated time-points uncover the peak of PREP1 binding at 12 h and PBX1/MEIS2 at 24 h. **g** Top: schematics of *Msgn1* locus and formation of the PBX/MEIS2/HOX-1 complex. Bottom: *Msgn1-P2* oligonucleotide sequence (50 bp in the middle of PBX1 ChIP-seq peak) containing WT (upper case) and mutated (lower case) PBX/HOX-1-binding site (red). **h** EMSA with in vitro-translated TALE proteins and *Msgn1-P2* oligonucleotide. The composition of each binding reaction is indicated. A ternary complex (TC) is formed only when PBX, MEIS2 and HOXA1 are mixed together (lines 4–6). TC is co-migrating with a complex present in MPCs nuclear extracts (line 8). The oligonucleotide with single-base substitutions in the PBX/HOX-binding site (*Msgn1-P2-M*) abrogates all complex formation (lines 12–14). LYS, reticulocyte lysate (lines 7 and 15). **i** ATAC-seq of *Msgn1* locus in WT and *Pbx-DKO* cells along differentiation reveals specific opening of PBX1 and LEF1-binding sites. ATAC-seq signals in WT mimic *Msgn1* transcriptional profile, with maximum accessibility and expression at 24 h. In *Pbx-DKO* cells, accessibility to *Msgn1* is reduced in the LEF1-binding and the PBX/HOX-1-binding regions. Threshold of vertical viewing range of data based on RPKM values is indicated in **c** and **i**. Data are mean ± s.e.m. (*n* = 2 biological replicates) in **b** and **f**.

gRNA-pX335 plasmids (gRNA 1: GTCCAGGCTCTGGTCCGTGA; gRNA 2: CCAA GTGAGTGCCCCCACCC; 100 ng each vector) and 1 µg linearised targeting vector using Lipofectamine 2000 (Thermo Fisher Scientific, 11668030), seeded at low density in EpiSC medium in 100 mm tissue-culture dishes (Corning, 430293) pre-coated with 15 µg/ml human Fibronectin, and selected for hygromycin resistance for 8–10 days. Single colonies were screened by diagnostic PCR, Southern blot and western blot analyses. To reduce side effects caused by CRISPR/Cas9 off-targets, we subsequently employed the High-fidelity CRISPR/Cas9 system (Cas9-HF1)[58] to target Exon1 of *Pbx1* in *Pbx2-KO* EpiSC lines. The gRNA used (GGCCATCCCGACCCCAGCGT) was cloned into a modified version of pX458 (a gift from Prof. F. Zhang; Addgene, 48138) containing the sequence encoding for Cas9-HF1 (a kind gift from Dr. F.V. Roske (Novo Nordisk, Denmark)). Cells were transfected with 300 ng gRNA-pX458 and after 48 h GFP[pos] cells were re-seeded at low density in EpiSC medium in Fibronectin-coated 100 mm tissue-culture dishes. After 6-8 days, single colonies were screened by diagnostic PCR and selected *Pbx-DKO* lines were expanded and validated by Western blot.

**Generation of *Pbx-DKO* lines by CRE-mediated recombination.** EpiSC lines were derived in EpiSC medium as previously described[56] by Dr. J. Martin Gonzalez (Core Facility for Transgenic Mice, Faculty of Health and Medical Sciences, University of Copenhagen, Copenhagen, Denmark) from mixed background *Pbx1[fl/fl];Pbx2[−/−]* mouse embryos. Additionally, *Yfp* (yellow fluorescent protein) cDNA was inserted at the *Rosa26* locus preceded by a *loxP*-flanked stop sequence to generate CRE recombination reporter lines. After isolation, cell lines were cultured as described above for a few passages, transfected with 200 ng pCAG-Cre (a gift from Prof. C. Cepko; Addgene, 13775) using Lipofectamine 2000 and re-seeded at low density in EpiSC medium in 100 mm tissue-culture dishes pre-coated with 15 µg/ml human Fibronectin. After 6–8 days, single YFP[pos] colonies were picked and screened by PCR. The selected lines were expanded and their genotype was further validated by western blot.

**Generation of *pMsgn1-mut* lines using the CRISPR/Cas9 system.** To generate the *pMsgn1-mut* lines, CRISPR/Cas9-mediated homology recombination was performed on 129 EpiSCs using a single-stranded DNA (ssDNA) donor template carrying four base substitutions that abrogate the recruitment of PBX to the *Msgn1* promoter, as assessed by EMSA (Fig. 6g–h). The gRNA designed for targeting the *Msgn1* regulatory region (shown in Supplementary Fig. 8a) was cloned into pX459 (a gift from Prof. F. Zhang; Addgene, 62988). Cells were co-transfected with 200 ng gRNA-pX459 and 500 pM ssDNA donor template using Lipofectamine 2000, re-seeded at low density in EpiSC medium in 100 mm tissue-culture dishes pre-coated with 15 µg/ml human Fibronectin and selected for neomycin resistance. Single colonies were screened by diagnostic PCR and restriction enzyme analysis. The selected lines were expanded and the gene editing was further validated by sequencing.

**EpiSC differentiation to PSM and derivatives.** All differentiation was conducted in serum-free, feeder-free and monolayer conditions in chemically defined N2B27-based media. Culture conditions followed a modified protocol from Loh et al.[59]. EpiSCs were seeded at ~6 × 10³ cells/cm² in six-well cell-culture plates pre-coated with 15 µg/ml human Fibronectin, and cultured in EpiSC medium for 24 h. EpiSCs were first differentiated towards APS using 30 ng/ml Activin A, 4 µM CHIR-99021, 20 ng/ml bFGF and 100 nM PIK-90 (APS medium) for 12 h, and then to PSM using 1 µM A-83-01, 3 µM CHIR-99021, 250 nM LDN193189 and 20 ng/ml bFGF (PSM medium) for 36 h. For PSM derivatives, cells were kept in PSM medium for additional 24 h, and then differentiated towards ventral somites/sclerotome using 5 nM SAG 21k and 1 µM C59 (sclerotome medium) or dorsal somites/dermo-myotome using 3 µM CHIR-99021 and 150 nM GDC-0449 (dermomyotome

medium) until day 5. A complete list of cytokines and signalling pathway inhibitors used for cell maintenance and differentiation is provided in Supplementary Table 1.

**Lentivirus-mediated exogenous expression of *Msgn1*.** The murine *Msgn1*-coding sequence was amplified from cDNA, subcloned into the VIRSP lentiviral vector between the restriction sites NheI and XhoI and verified by sequencing. In order to produce the lentivirus carrying the VIRSP-*Msgn1* vector, HEK293T cells were seeded at ~9 × 10⁴ cells/cm² in 100 mm tissue-culture dishes in Dulbecco's Modified Eagle's Medium (DMEM, Thermo Fisher Scientific, 10566016) + 10% (v/v) Fetal Bovine Serum (FBS, Gibco, 10270-106). The following day, the medium was changed to N2B27, 1 h prior to transfection. Cells were transfected with 15 µg VIRSP-*Msgn1*, 10 µg psPAX2 (a gift from Prof. D. Trono; Addgene, 12260), 5 µg pCMV-VSV-G (a gift from Prof. R.A. Weinberg; Addgene, 8454) and 90 µl PEI (1 mg/ml, Sigma Aldrich, 765090). The supernatant containing the lentivirus was collected after 72 h and concentrated using ultracentrifugation filter units (Merck Millipore, UFC910008) as per manufacturer's instructions. The concentrated supernatant was stored at −80 °C. To perform the rescue experiment, WT and *Pbx-DKO* 129 EpiSCs were seeded in six-well cell-culture plates pre-coated with 15 µg/ml human Fibronectin. In all, 3.5 µl of lentivirus were added 4 h after seeding, and then differentiation to PSM was carried out as described above. Total RNA was extracted from samples harvested after 48 h of differentiation and the rescue efficiency was assessed by RT-qPCR as compared to WT EpiSCs with and without added virus, as well as to *Pbx-DKO* cells without added virus.

**Immunofluorescence staining of adherent cells.** EpiSCs were seeded at ~4 × 10³ cells/cm² in EpiSC medium in eight-well microscopy slides (Ibidi, 80826) pre-coated with 15 µg/ml human Fibronectin, and differentiated to PSM as described above. At desired time-points, cells were briefly washed twice in PBS (Sigma, D8537) and fixed in 4% (w/v) paraformaldehyde diluted in PBS from 16% methanol-free formaldehyde stock (Pierce Biotechnologies, 28906) for 15 min at RT. Fixed cells were then permeabilised in PBS containing 0.1% (v/v) Triton X-100 (PBST) for 10 min at RT and blocked in 3% (v/v) Donkey serum in PBST containing 1% (w/v) BSA (blocking solution) for 15 min at RT. Primary and secondary antibody incubations were done in blocking solution overnight at 4 °C and for 2 h at RT, respectively. Antibody incubations were followed by three quick washes in PBST followed by two washes in PBST for 15 min at RT. After the final incubation, cells were stored in PBST containing 1 µg/ml DAPI at 4 °C, packed in Parafilm to prevent evaporation. Cells were always kept in the dark after incubation with secondary antibodies. Confocal imaging was performed using a Leica TCS SP8 microscope and images were processed using the Imaris[TM] software (v9.3.0) with default settings. The antibodies used are listed in Supplementary Table 2.

**Imaging of embryo morphology.** Embryos were dissected in cold PBS, fixed in 4% (w/v) paraformaldehyde for 30 min at RT and washed twice in cold PBS. Images were acquired using a Zeiss AxioZoom.V16 microscope at the Core Facility for Integrated Microscopy (University of Copenhagen, Copenhagen, Denmark).

**Immunofluorescence staining of embryos.** Embryos were dissected in cold PBS and fixed in 4% (w/v) paraformaldehyde for 30 min at RT. Fixed embryos were rinsed in cold PBS containing 3 mg/ml PVP (PBS/PVP), permeabilised in PBS/PVP + 1% (v/v) Triton X-100 for 10 min at RT and blocked in CAS-Block (Thermo Fisher Scientific, 008120) overnight at 4 °C. Incubation with the primary antibodies listed in Supplementary Table 3 was carried out in CAS-Block for 48 h at 4 °C. Embryos were quickly washed three times with CAS-Block at RT and then overnight at 4 °C. Incubation with secondary antibodies was done in CAS-Block for 24 h at 4 °C. Embryos were quickly washed three times with CAS-Block at RT and

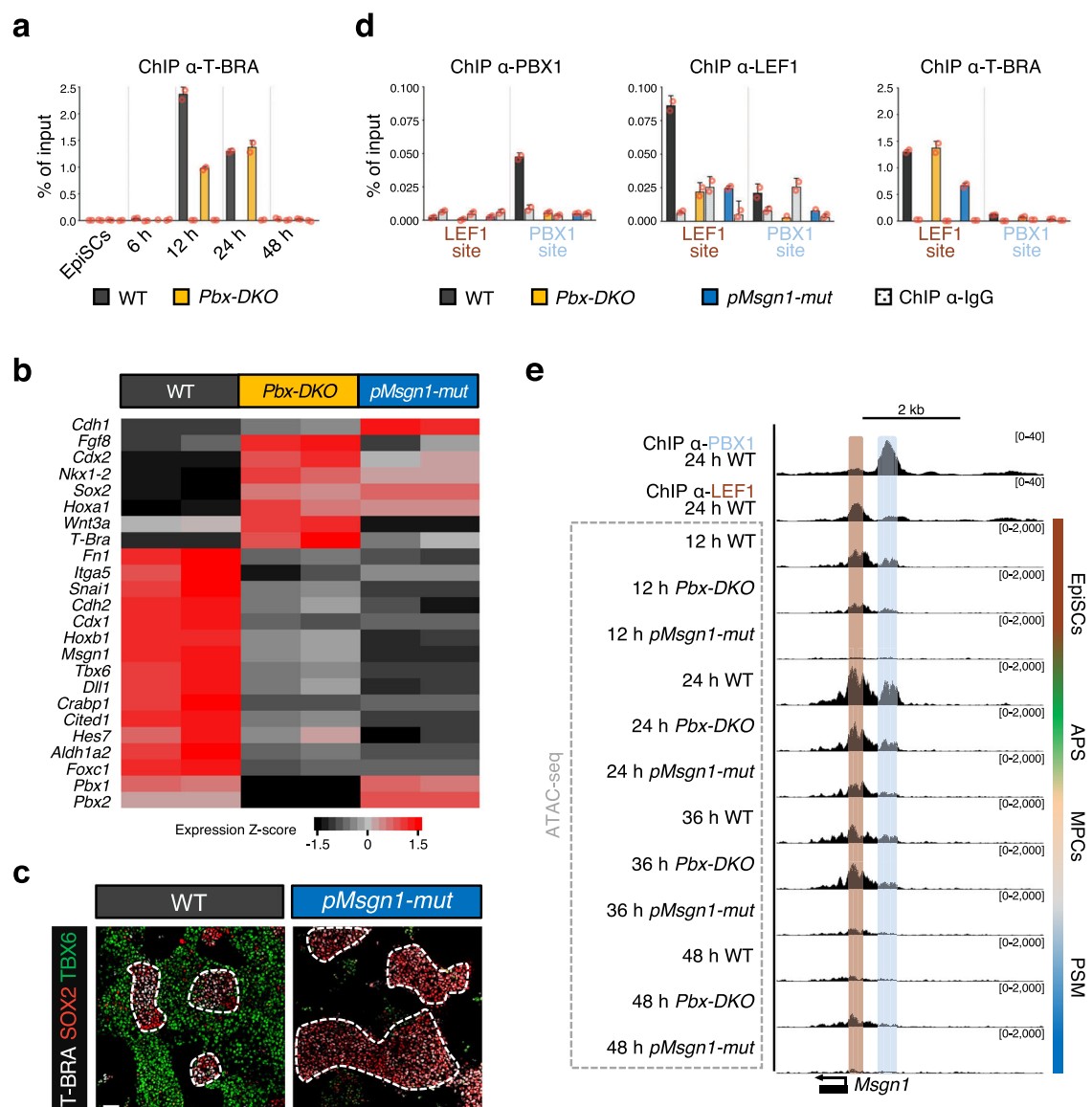

**Fig. 7 Combined pioneering activity of T-BRA and TALE proteins remodels the chromatin, making it accessible at *Msgn1* regulatory regions, thereby directing WNT transcriptional response. a** ChIP-qPCR analysis showing specific recruitment of T-BRA to the LEF1-binding site of the *Msgn1* regulatory region at 12 and 24 h. T-BRA binding is maintained in *Pbx-DKO* cells (yellow). **b** Heatmap displaying relative expression of selected differentially regulated genes in WT (black), *Pbx-DKO* (yellow) and *pMsgn1-mut* cells (blue) at 24 h (RNA-seq, P ≤ 0.05, fold-change ≥1.5). *pMsgn1-mut* allele carries point mutations in the PBX/HOX-1-binding site. Representative PSM and NMP markers are indicated. n = 2 biological replicates. Significance was assessed by DESeq2 on the basis of two-sided Wald test with Benjamini–Hochberg adjusted P-values. **c** Representative immunofluorescence staining for T-BRA (white), SOX2 (red) and TBX6 (green) at 48 h of differentiation reveals expansion of NMPs (T-BRA[pos];SOX2[pos], dashed white lines) and reduction of PSM (TBX6[pos]) in *pMsgn1-mut* cells. Scale bar: 50 μm. **d** ChIP-qPCR analysis on the *Msgn1* promoter displays PBX1 recruitment at the PBX/HOX-1-binding site in WT (black), but not in *Pbx-DKO* (yellow) nor in *pMsgn1-mut* cells (blue). Similarly, ChIP-qPCR analyses with LEF1 and T-BRA antibodies highlight specific binding of LEF1 and T-BRA to the LEF1-binding region in WT cells. While T-BRA is recruited to the *Msgn1* promoter in *Pbx-DKO* and *pMsgn1-mut* cells, LEF1 binding is lost in both. Data are mean ± s.e.m. (n = 2 biological replicates) in **a** and **d**. **e** Chromatin accessibility analysis (ATAC-seq) of the *Msgn1* locus in WT, *Pbx-DKO* and *pMsgn1-mut* cells along PSM differentiation displays specific open regions overlapping with the PBX (light blue) and LEF1-binding sites (brown) in WT cells at 12 and 24 h. In *Pbx-DKO* cells, accessibility is strongly reduced at the PBX/HOX-1-binding region. Additionally, accessibility of the *Msgn1* locus is severely impaired in both LEF1 and PBX/HOX-1-binding regions in *pMsgn1-mut* cells, confirming the role of the PBX complexes as modulators of chromatin accessibility on WNT-responsive elements. Threshold of vertical viewing range of data based on RPKM values is specified.

then overnight at 4 °C. Following incubation with CAS-Block containing 1 µg/ml DAPI for 1 h at RT, embryos were washed three times with PBS + 0.5% (v/v) Triton X-100 for 15 min at RT and dehydrated in ascending series to 100% methanol. To reduce light scattering, optical clearing was performed by submerging the stained embryos in a 1:2 (v/v) solution of benzyl alcohol:benzyl benzoate (BABB). Images were acquired using a Leica TCS SP8 confocal microscope. Image processing and 3D reconstruction were performed using the Imaris™ software (v9.3.0). In Figs. 2c, d and 3i and Supplementary Fig. 2b, only cells expressing high levels of either T-BRA, SOX2 or TBX6 were included in the scoring.

**In situ hybridisation.** Embryos were dissected in cold PBS and fixed in 4% (w/v) paraformaldehyde for 2 h at RT, washed twice in cold PBS with 0.1% Tween-20 and dehydrated in ascending series to 100% methanol. At least three embryos of the same genotype were used. Whole-mount in situ hybridisation was performed following the protocol used in Prof. E. De Robertis' laboratory (http://www.hhmi.ucla.edu/derobertis/protocol_page/Pdfs/Mouse%20protocols/Whole%20mount%20in%20situ%20hybridization%20on%20mouse%20embryos.pdf). The antisense riboprobes to *Msgn1*[60] and *Wnt3a*[61] were a kind gift from Prof. A.K. Hadjantonakis (Memorial Sloan Kettering Cancer Center, New York, United States).

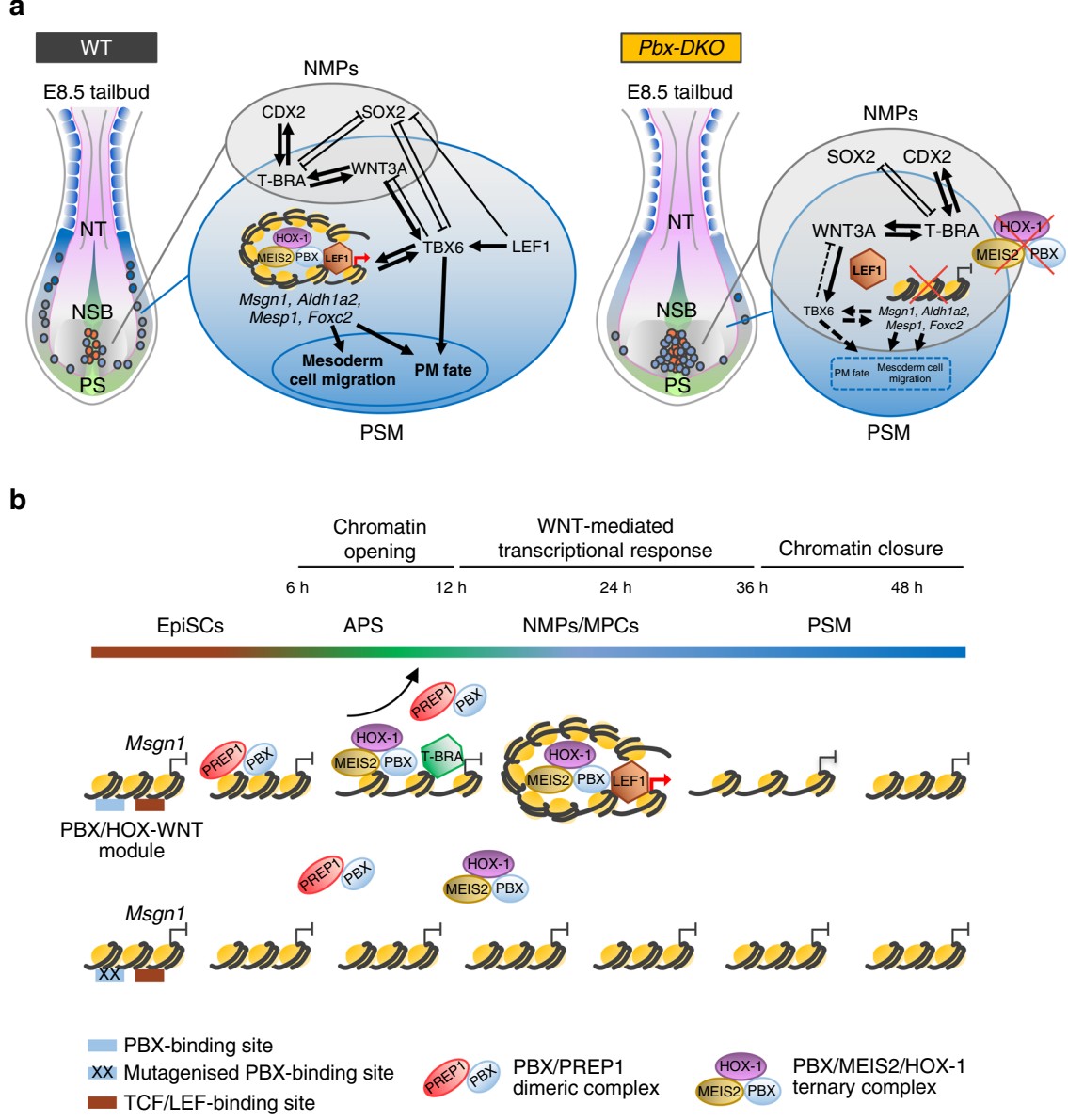

**Fig. 8 TALE complexes promote WNT-mediated transcriptional response in paraxial mesoderm. a** Schematics of the molecular networks at play in the E8.5 embryonic tailbud. Left: cartoon showing NMPs (red dots) transitioning to MPCs (light blue dots) and migrating to the PSM (blue) in WT embryos. GRNs governing the balance between NMP expansion in the CLE (grey) and PSM differentiation (blue) are illustrated. The TALE complexes and LEF1 bind cooperatively to the regulatory regions of PM genes, like *Msgn1, Aldh1a2, Mesp1* and *Foxc2*, promoting their expression. The activation of the PM programme ultimately controls migration of MPCs from the progenitor zone to the PSM and the acquisition of PM fate. Right: PBX loss (yellow) results in reduced mobility of the MPCs and accumulation of NMPs-MPCs in the progenitor zone, leading to enlarged tailbuds and reduced PSM formation. Failure of TALE complex assembly on the regulatory regions of PM genes impairs LEF1 recruitment and activation of the GRN promoting PSM formation. In contrast, the molecular circuits sustaining NMP expansion are maintained by positive feedback loops. Of note, the observed PBX-LEF1 binding to the *N1* repressive element of the *Sox2* regulatory region suggests that PBX and LEF1 cooperative activity could repress *Sox2* expression (Fig. 5e) and NMP maintenance. Thus, PBX proteins are required for the transition of NMPs to PSM, playing essential and previously unappreciated roles in the early stages of somitogenesis. CLE caudal lateral epiblast, NSB node-streak border, PS primitive streak, PSM pre-somitic mesoderm, NT neural tube. **b** Model of the transcriptional activation of *Msgn1*. *Msgn1* regulatory region becomes accessible at 12 h following the collaborative pioneering activity of T-BRA and the TALE complexes. Low-affinity PBX/PREP1 dimers are recruited first and could serve as HOX-1 attractors, or assist the loading of the high-affinity PBX/MEIS2/HOX-1 ternary complex, whose cooperative binding displaces nucleosomes and generates the suitable chromatin context for LEF1 recruitment. Single-base substitutions at the PBX/HOX-1-binding site abrogate accessibility of both PBX/HOX-1 and LEF1-binding regions, emphasising the role of TALE complexes as modulators of chromatin accessibility on WNT-responsive elements.

**Time-lapse videos of PSM differentiation**. EpiSCs were seeded at ~$6 \times 10^3$ cells/cm$^2$ in six-well cell-culture plates pre-coated with 15 μg/ml human Fibronectin, and differentiated to PSM as described above. Differentiation was carried out in the IncuCyte S3 Live-Cell Analysis System (Essen Bioscience) at the High-Throughput Cell-based Screens Facility (Biotech Research & Innovation Centre, University of Copenhagen,

Copenhagen, Denmark), and images were acquired every h for 3 days. Time-lapse movies with all acquired pictures were further recreated using iMovie (version 10.1.14).

**Western blot**. In all, 80% confluent cells were washed once with PBS and then lysed directly on the plate by addition of 1x Laemmli sample buffer (2% (w/v) SDS,

10% (v/v) glycerol and 120 mM Tris-HCl pH 6.8) to the culture dish (20 μl/cm²), followed by extensive scraping using a pipette tip. Samples were vortexed, centrifuged briefly and then stored at −20 °C until use. Before loading, samples were boiled for 5 min at 80 °C, sonicated on ice for 10 s at 20% power using a Branson Digital Sonifier Model S-450D (Marshall Scientific), spun for 10 min at 14,000 × g to clear the lysates and then stored at −20 °C until use. Protein concentration was estimated using NanoDrop (Thermo Fisher Scientific) A280 measurements and equal amounts of proteins were diluted in 60 μl of 1x Laemmli sample buffer per sample. After addition of 6 μl of 1 M DTT containing 0.01% (w/v) bromophenol blue, 50 μl of sample were loaded per lane on NuPAGE Novex 4–12% Bis-Tris Protein Gels (Thermo Fisher Scientific, NP0321BOX). Electrophoresis was performed in 1x NuPAGE MES SDS running buffer (Thermo Fisher Scientific, NP0002) at 150 V for 1 h. Resolved proteins were transferred to Amersham Protran Nitrocellulose Blotting membrane 0.45 μm (GE Healthcare, 10600003) at 400 mA for 90 min on ice in 1x NuPAGE transfer buffer (Thermo Fisher Scientific, NP0006). After a quick wash in PBS containing 0.1% (v/v) Tween-20 (PBST), membranes were blocked for 1 h at RT in PBST containing 10% Skim milk powder. All subsequent antibody incubations were performed in PBST containing 5% (w/v) BSA and followed by three washes in PBST for 10 min. The following primary antibodies were used: polyclonal Rabbit anti-PBX1 (CST, 4342, 1:1,000), polyclonal Rabbit anti-PBX2 (Santa Cruz Biotechnology, sc-890, 1:3,000) and monoclonal Mouse anti-H3 (abcam, ab10799, 1:30,000). The secondary antibodies used include: Donkey anti-Rabbit HRP (Thermo Fisher Scientific, A16023, 1:10,000) and Donkey anti-Mouse HRP (Thermo Fisher Scientific, A16011, 1:10,000). For chemiluminescence detection, the membrane was incubated with Amersham ECL Prime Western Blotting Detection reagent (GE Healthcare, RPN2236) and imaged on a ChemiDoc MP (Bio-Rad). For reprobing with different primary antibodies, the membrane was soaked in Restore PLUS Western Blot Stripping Buffer (Thermo Fisher Scientific, 46430) two times for 10 min at RT to inactivate horseradish peroxidase (HRP). To ensure comparable densitometry results, gel electrophoresis, protein transfer and antibody staining were performed in parallel for all samples used to generate each western blot figure panel.

**Electrophoretic Mobility Shift Assay**. EMSAs were performed as described[62], using in vitro-translated proteins or nuclear extracts purified from cells. Sequences of the oligonucleotides spanning the PBX/HOX-1-binding site on the *Msgn1* promoter used in EMSA are described in Supplementary Fig. 7a. EMSAs with in vitro-translated proteins were performed using 5 μg of nuclear extract or 2 μl of reticulocyte lysate containing the desired combination of proteins mixed with 18 μl of PPH binding buffer (10 mM Tris-Cl pH 7.5, 75 mM NaCl, 1 mM EDTA, 6% (v/v) glycerol, 3 mM spermidine, 1 mM DTT, 0.5 mM PMSF, 1 μg poly (dIdC), 30,000 c.p.m. ³²P-labelled oligonucleotide) to a total volume of 20 μl. After 30 min of incubation on ice the reactions were separated by 5% acrylamide gel in 0.5x TBE. For the competition assays 50, 100 or 200-molar excess of unlabelled competitor oligonucleotide were added to the binding reaction 10 min before the labelled probe. Acrylamide gels were pre-run at 300 V for 90 min. Samples were separated on gel at 150 V for 2 h. Gels were dried on gel dryer, exposed overnight and imaged on a Typhoon 9400 imager (GE Healthcare). Based on the known molecular weight of the in vitro-translated proteins, the expected sizes of dimeric and ternary complexes are as follows: Fig. 6h: PBX1A/MEIS2 = ~114 kDa, PBX1B/MEIS2 = ~104 kDa, PBX2/MEIS2 = ~114 kDa, PBX1A/MEIS2/HOXA1 = ~149 kDa, PBX1B/MEIS2/HOXA1 = ~139 kDa, PBX2/MEIS2/HOXA1 = ~149 kDa; Supplementary Fig. 7c: PBX1A/MEIS2 = ~114 kDa, PBX1B/MEIS2 = ~104 kDa, PBX2/MEIS2 = ~114 kDa, PBX3/MEIS2 = ~114 kDa, PBX1A/MEIS2/HOXA1 = ~149 kDa, PBX1B/MEIS2/HOXA1 = ~139 kDa, PBX2/MEIS2/HOXA1 = ~149 kDa, PBX3/MEIS2/HOXA1 = ~149 kDa, PBX1A/MEIS2/HOXB1 = ~146 kDa, PBX1B/MEIS2/HOXB1 = ~136 kDa, PBX2/MEIS2/HOXB1 = ~146 kDa, PBX3/MEIS2/HOXB1 = ~146 kDa; Supplementary Fig. 7d: PBX1A/PREP1 = ~114 kDa, PBX1B/PREP1 = ~104 kDa, PBX2/PREP1 = ~114 kDa, PBX3/PREP1 = ~114 kDa, PBX1A/MEIS2 = ~114 kDa, PBX1B/MEIS2 = ~104 kDa, PBX2/MEIS2 = ~114 kDa, PBX3/MEIS2 = ~114 kDa; Supplementary Fig. 7e: PBX1A/PREP1 = ~114 kDa, PBX1A/PREP1/HOXA1 = ~149 kDa, PBX1A/MEIS2 = ~114 kDa, PBX1A/MEIS2/HOXA1 = ~149 kDa; Supplementary Fig. 7f: PBX3/PREP1 = ~114 kDa, PBX1A/MEIS2/HOXB1 = ~146 kDa; and Supplementary Fig. 7g: PBX1A/PREP1 = ~114 kDa, PBX1A/MEIS2/HOXB1 = ~146 kDa.

**Oligonucleotides preparation**. Oligonucleotides were resuspended in TE pH 8.0. Double-stranded oligonucleotides were prepared by mixing equal molar ratios of single-strand oligonucleotides. The mixture was heated at 90 °C for 10 min and slowly cooled at RT. Perfect oligo annealing was checked by electrophoresis on 15% acrylamide gel. For oligos longer than 40 bp, double-strand oligonucleotides purification was performed using preparative acrylamide 15% in 0.5x TBE. The oligonucleotides were eluted from the gel in 500 μl TEN (10 mM Tris, 1 μM EDTA, 300 mM NaCl, pH 8.0) with overnight shaking. The oligonucleotides were recovered by ethanol precipitation. The pellet was washed, dried and resuspended in 50 μl TE pH 8.0.

**Oligonucleotides 5′ end labelling**. In total, 0.5 pmoles of double-stranded oligonucleotides were incubated with 1 μl of 10x PNK buffer 5 Units Polynucleotide Kinase (Roche, 10174645001) and [γ-³²P]ATP 3000 Ci/mMole (PerkinElmer,

BLU502A250UC). The reaction was incubated at 37 °C for 40 min, diluted to 50 μl with TE pH 8.0 and purified on a packed Sepharose G-25 column (Roche, 11273949001).

**Nuclear extract preparation**. Nuclear extracts (NE) for EMSA were prepared from in vitro-differentiated paraxial mesoderm cells. Cells were resuspended in 200 μl lysis buffer (10 mM HEPES pH 7.9, 30 mM KCl, 1.5 mM MgCl₂, 1 mM DTT, supplemented with 1x Protease inhibitor cocktail (Roche, 11836170001)), kept on ice for 10 min and supplemented with addition of Triton X-100 to a final concentration of 0.1%. The nuclei were collected by centrifugation at 4 °C. Nuclear extracts were prepared by resuspending the nuclear pellet in 50 μl nuclear buffer (20 mM HEPES pH 7.9, 25% (v/v) glycerol, 0.42 M NaCl, 1.5 mM MgCl₂, 1 mM DTT, supplemented with 1x Protease inhibitor cocktail). NE preparations were incubated at 4 °C with rotation for 30 min and cleared by centrifugation. Finally, the extracts were quantified by NanoDrop, aliquoted and stored at −80 °C.

**Cell-free protein synthesis**. The cDNAs were transcribed and translated using the couple T7-TNT reticulocytes lysate system (Promega, L1170) according to the manufacturer's instructions. All cDNAs of the proteins used in this study (PBX1A, PBX1B, PBX2, PBX3A, PREP1, PREP2, MEIS2, HOXA1 and HOXB1) were cloned in vectors under the control of T7 promoter. The PBX3A (MC202386) and PREP2 (MC204820) constructs were purchased from OriGene Technologies. All the other plasmids were described in Ferretti et al.[62].

**scRNA-seq procedure**. scRNA-seq was performed using the Massively Parallel Single-Cell RNA-sequencing technology (MARS-seq[63]). For in vitro experiments, EpiSCs were seeded at ~6 × 10³ cells/cm² in six-well cell-culture plates pre-coated with 15 μg/ml human Fibronectin, and differentiated to PSM as described above. At desired time-points, cells were dissociated using Accutase (Thermo Fisher Scientific, 00-4555-56) for 3 min at 37 °C and counted. 500,000 cells per condition were washed with ice-cold FACS buffer (10% (v/v) FBS in PBS) and resuspended into 1 ml ice-cold FACS buffer containing 1 μg/ml DAPI. For in vivo experiments, tailbuds were dissected from E8.5/E9.0 embryos and dissociated into 300 μl warmed Trypsin 0.05% (w/v) EDTA for 10 min at 37 °C. Digestion was stopped by addition of 600 μl ice-cold FACS buffer followed by centrifugation at 350 × g for 3 min. Cells were washed with 500 μl ice-cold FACS buffer, centrifuged again at 350 × g for 3 min and resuspended into 250 μl ice-cold FACS buffer containing 1 μg/ml DAPI. Singlecells from in vitro-differentiated cells or embryonic tailbuds were sorted into Eppendorf Polypropylene U-shaped 384-well Twin Tec PCR Microplates (Thermo Fisher Scientific, 10573035), containing 2 μl of lysis solution (0.2% (v/v) Triton X-100) supplemented with 0.4 U/μl RNasin Ribonuclease Inhibitor (Promega, N2515) and 400 nM indexed RT primer from group 1 (1–96 barcodes) or group 2 (97-192 barcodes)[63]. Additionally, 71 WT EpiSCs were sorted into each plate, as spike-in control for batch-effect correction. Capture plates were prepared on the Bravo automated liquid handling robot station (Agilent) using 384-filtered tips (Agilent, 19133-142). Index sorting was performed using a FACS Aria III cell sorter (BD Biosciences) at the DanStem Flow Cytometry Platform (University of Copenhagen, Copenhagen, Denmark), sorting live cells as singlets. Immediately after sorting, plates were spun down, snap-frozen on dry ice and stored at −80 °C until further processing. Semi-automated library preparation was performed, using 10–12 total cycles of PCR amplification and AMPure XP beads (Beckman Coulter Life Sciences, A63881) for purification[63]. DNA concentration was measured with a Qubit Fluorometer (Thermo Fisher Scientific, Q32854) and fragment size was determined with a Fragment analyzer (Agilent). Libraries were paired-end sequenced on a Next-Seq 500 Sequencer (Illumina) at the DanStem Genomics Platform (University of Copenhagen, Copenhagen, Denmark). Between 1,146 and 1,528 cells were sequenced per lane.

**RNA-seq procedure**. EpiSCs were seeded at ~6 × 10³ cells/cm² in six-well cell-culture plates pre-coated with 15 μg/ml human Fibronectin, and differentiated to PSM as described above. At desired time-points, cells were lysed and total RNA was extracted using the RNeasy Mini Kit according to the manufacturer's instructions (Qiagen, 74106). RNA concentration was quantified by NanoDrop and quality was verified with a Fragment analyzer. Library preparation was carried out using 0.5 μg of RNA with NEBNext Ultra II RNA library Prep Kit (New England BioLabs, E7770S) as per manufacturer's instructions. Libraries were amplified for 4 total PCR cycles and purified with AMPure XP beads. DNA concentration was measured with a Qubit Fluorometer and fragment size was determined with a Fragment analyzer. All samples were sequenced in biological triplicates on a Next-Seq 500 Sequencer (Illumina) at the DanStem Genomics Platform (University of Copenhagen, Copenhagen, Denmark).

**ATAC-seq procedure**. ATAC-seq was performed on 50,000 cells per replicate as described in Buenrostro et al.[64] (with modifications based on Corces et al.[65]), on EpiSCs and PSM-differentiated cell populations at desired time-points. Libraries were generated using the Ad1_noMX and Ad2.1–2.16 barcoded primers[64] and amplified for 10 total PCR cycles. Libraries were purified with AMPure XP beads to remove contaminating primer dimers and fragments >1,000 bp. Library quality was assessed using the Fragment analyzer and quantitated by Qubit assay. The libraries were sequenced with 50 bp

paired-end reads on a Next-Seq 500 Sequencer (Illumina) at the DanStem Genomics Platform (University of Copenhagen, Copenhagen, Denmark).

**ChIP-seq procedure**. ChIP was performed with a protocol modified from that of Kolasinska-Zwierz et al.[66]. EpiSCs were seeded at ~2 × 10³ cells/cm² in EpiSC medium in 150 mm tissue-culture dishes (Corning, 430599) pre-coated with 15 μg/ml human Fibronectin, and differentiated to PSM as described above. At desired time-points, cells were washed with PBS and fixed directly on the plate by the addition of 20 ml of PBS containing 1% (v/v) Formaldehyde Fixative Ultrapure EM Grade (Rockland, KHF001) for 8–10 min at RT. Fixation was stopped by 5 min incubation with 125 mM glycine at RT. After two washes with cold PBS, fixed cells were harvested in lysis buffer (50 mM Tris-HCl pH 8.1, 100 mM NaCl, 5 mM EDTA, 0.2% (w/v) NaN₃, 0.5% (w/v) SDS, supplemented with 1x Protease inhibitor cocktail and 1 mM PMSF) by extensive scraping using a pipette tip. Following centrifugation at 1,000 × g for 5 min at 4 °C, the pellet was resuspended in 1 ml of FA buffer (50 mM HEPES/KOH pH 7.5, 1 mM EDTA, 1% (v/v) Triton X-100, 0.1% sodium deoxycholate, 150 mM NaCl, supplemented with 1x Protease inhibitor cocktail) per 0.5 ml of pellet. Chromatin was disrupted by sonication using a chilled Bioruptor Plus sonicator UCD-300 (Diagenode) for 8–12 pulses (cycles of 30 s on/45 s off, high setting at 4 °C) to obtain fragments of 200–500 bp in size. The extract was spun for 30 min at 20,000 × g at 4 °C, and the soluble fraction was aliquoted and stored at −80 °C until use. For each IP, 20 μg of chromatin were incubated overnight in 1 ml of FA buffer containing 1% N-lauroylsarkosine sodium salt with the antibodies listed in Supplementary Table 4. 10% of the IP volume of the chromatin was retained as an input reference. After overnight rotation at 4 °C, 30 μl of washed and blocked magnetic Protein A (Thermo Fisher Scientific, 10006D) or Protein G Dynabeads (Thermo Fisher Scientific, 10003D) were added and the incubation continued for 2 additional h. Beads were extensively washed at RT[66]. DNA was eluted twice with 57 μl of elution buffer (1% (w/v) SDS in TE pH 8.0 with 250 mM NaCl) at 65 °C, 15 min each time. Eluted DNA was incubated with 20 μg of RNase A for 30 min at 37 °C and then with 20 μg of Proteinase K for 1 h at 55 °C. Input DNA was also diluted in 114 μl of elution buffer and treated as ChIP samples. Crosslinks were reversed at 65 °C overnight and DNA was purified using the QIAGEN MinElute PCR Purification Kit (QIAGEN, 28006) according to the manufacturer's instructions (Qiagen). DNA concentration was measured with a Qubit Fluorometer and fragment size was determined with a Fragment analyzer. For ChIP-seq, libraries were prepared using the NEBNext Ultra II DNA Library Prep Kit (New England BioLabs, E7645S) as per manufacturer's instructions. Libraries were amplified for 5–7 total PCR cycles, purified with AMPure XP beads and sequenced on a Next-Seq 500 Sequencer (Illumina) at the DanStem Genomics Platform (University of Copenhagen, Copenhagen, Denmark). For ChIP-qPCR, immunoprecipitated DNA and input were quantified by real-time qPCR as described below. The following primers and probes (Universal Probe Library, UPL, Roche) were used: for *Msgn1* PBX/HOX-1-binding site, Fw, GTGCAGAAA-TATCCCCTGCT; Rv, TGTCTTCTGGTGCTGTTTGC; Probe, 95; for *Msgn1* LEF1-binding site, Fw, GCATCTTCCAGCTAAAATGTCTTT; Rv, GCAGCCT-TAATTGGTTTAGTTACAA; Probe, 75. The measures were normalised to the input. All reactions were performed in duplicates.

**Real-time quantitative PCR**. Total RNA was extracted from 0.5–1 × 10⁶ cells using the RNeasy Mini Kit according to the manufacturer's instructions. First-strand synthesis was performed on 1 μg of total RNA using random hexamers (Thermo Fisher Scientific, N8080127) and SuperScript III reverse transcriptase (Thermo Fisher Scientific, 18080-044) according to the manufacturer's instructions. A mix of concentrated cDNA from all samples was used to generate standard curves. Amplification was detected using the Universal Probe Library system on a LightCycler480 Real-Time PCR System (Roche). The measures were normalised to *Tbp* and *Sdha* reference genes. All reactions were performed in duplicates. See Supplementary Table 5 for a list of primers and probes used.

**scRNA-seq data analysis**. R1 and R2 fastq files were generated using bcl2fastq (v2.19.1), and the pooling and well information were extracted from the sequence using umis (v1.0.3) [https://github.com/vals/umis] into a unique fastq file. The reads were then filtered based on the pooling barcodes with 1 mismatch allowed. The poly-Ts at the end of the reads were trimmed using Cutadapt (v1.18)[67]. The reads were mapped to the mouse genome (GRCm38/mm10 together with ERCC92) using HISAT2 (v2.1.0), the alignments were processed with Samtools (v1.7)[68], and the reads were counted with featureCounts (Subread (v1.5.3))[69] using Ensembl v93, and the umis using UMI_tools (v1.0.0)[70]. Expression data were analysed using Seurat (v3.1)[71,72]. Data filtering, normalisation and scaling were performed using the standard pre-processing workflow[71]. Integration of different datasets was performed as described in Stuart et al.[72]. Spiked-in EpiSCs were used as a reference to correct the batch-effect between integrated datasets. Marker genes of each cell cluster were outputted for GO-term analysis to define the cell type. The Monocle package (v2.16.0)[73] was used to perform pseudotime analyses. Single-cell trajectories were constructed in a semi-supervised manner, by defining NMPs as cells co-expressing moderate levels of *T-Bra/Sox2*, MPCs as co-expressing high levels of *T-Bra/Tbx6* and PSM as cells co-expressing high levels of *Tbx6/Meox1*.

**RNA-seq data analysis**. Fastq files were aligned to the GRCm38/mm10 genome using STAR (v2.5.3a)[74]. Transcript expression levels were estimated with the–quantMode GeneCounts option and GRCm38p5.vM15 annotations. FastQC (v0.11.7) (http://www.bioinformatics.babraham.ac.uk/projects/fastqc) was used for QC metrics, and MultiQC (v1.7)[75] for reporting. Data analysis was then performed with R/Bioconductor (v4.0.5) (https://www.R-project.org)[76]. Differentially expressed genes were identified using DESeq2 (v1.24.0)[77], which applies negative binomial generalised linear model fitting and Wald statistics on the count data. The results were filtered for Benjamini–Hochberg adjusted $P ≤ 0.05$ and fold-change ≥1.5, as indicated in the figure legends.

**GO-term analysis**. Biological pathway enrichments were determined using Panther (v15.0)[78]. Statistical significance was assessed with Binomial test with Bonferroni correction for multiple testing ($P ≤ 0.05$). Only the top12 terms ordered by Fold enrichment were considered and ranked by $P$-value.

**ATAC-seq data analysis**. Reads were trimmed with Cutadapt (v1.18)[79] (Cutadapt -a CTGTCTCTTATA -A CTGTCTCTTATA -q 20 -m 5), mapped to the GRCm38/mm10 mouse reference genome with Bowtie2 (v2.3.4.1)[80]. Mitochondrial reads and reads overlapping blacklisted regions (i.e. regions that are considered artefacts of ATAC-seq and other chromatin assays in mm10, https://github.com/MayurDivate/GUAVA/blob/master/lib/blacklists/JDB_blacklist.mm10.bed and http://mitra.stanford.edu/kundaje/akundaje/release/blacklists/mm10-mouse) were removed using bedtools (v2.27.1)[81], intersect and only uniquely mapped reads were kept using Samtools (v1.9)[82]. Bam files were sorted, deduplicated with Picard (v2.9.1) (http://broadinstitute.github.io/picard/) and indexed. Bigwig files were generated with bamCoverage (v3.2.0) from deepTools[83] and the RPGC option. MACS (v2.1.1.20160309) was used to call peaks[84].

**ChIP-seq data analysis**. Reads were mapped to the GRCm38/mm10 reference genome with Bowtie2 (v2.3.4.1)[80] and only uniquely mapped reads were kept using Samtools (v1.9)[82]. Bam files were sorted, deduplicated and indexed. Bigwig files were generated with bamCoverage (v3.2.0) from deepTools[83] and the RPGC option. The effective genome size was taken from the deepTools documentation. MACS (v2.1.1.20160309) was used to call peaks[84]. Heatmaps and peak profile were generated from the bigwig files with normalised counts to RPKM using deepTools (v3.2.0)[82]. The relative distance between peaks was calculated using the *closest* function in bedtools (v2.27.1)[81].

**De novo motif enrichment analysis**. Homer (v4.1)[85] de novo motif enrichment analysis was performed on bed files generated by MACS (v2.1.1.20160309)[84] peak caller on the GRCm38/mm10 reference genome with default settings.

**Venn diagrams**. Proportional Venn diagrams to show unique and intersecting elements of gene lists were generated using the web tool available at https://www.biovenn.nl.

**Statistics and reproducibility**. The sample size was estimated based on previous preliminary experiments. No statistical method was used to pre-determine sample size. Experiments and quantifications were not done in a blinded fashion. Unless otherwise noted, each immunofluorescence staining was repeated at least three times independently with similar results. Each western blot and EMSA were performed at least twice independently with similar results. Data are presented in box plots as median ± interquartile range as indicated in the figure legends, when $n ≥ 10$. Data are presented in histograms when $n < 10$, and individual data points are plotted. Information about $P$-values and statistical tests used in this article can be found in the figure legends. Statistics were derived using GraphPad Prism (v8.4.3) (GraphPad Software, San Diego, CA, USA). The raw data underlying all reported averages in graphs and charts, and the uncropped scans of western blots and EMSAs are provided in the Source data file.

**Reporting summary**. Further information on experimental design is available in the Nature Research Reporting Summary linked to this paper.

## Data availability

All data generated during this study are included in this article and its Supplementary Information files. scRNA-seq, bulk RNA-seq, ATAC-seq and ChIP-seq data that support the findings of this study have been deposited in the ArrayExpress database at EMBL-EBI (www.ebi.ac.uk/arrayexpress) under the following accession numbers: scRNA-seq of embryonic tailbuds, E-MTAB-9785; scRNA-seq of in vitro cells, E-MTAB-9774; bulk RNA-seq, E-MTAB-9773; ATAC-seq, E-MTAB-9776; and ChIP-seq, E-MTAB-9775. Source data are provided with this paper.

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

## Acknowledgements

We thank the staff of the DanStem Core Facilities: H.M. Neil, M. Michaut, G. De la Cruz, P. van Dieken, J.M. Bulkescher, A. Shrestha and A. Meligkova. We are grateful to M.M. Rothová for sharing the wild-type 129 EpiSC line, L. Selleri for providing the *Pbx* mutant alleles and the Core Facility for Transgenic Mice at the University of Copenhagen, especially J. Martin Gonzalez for derivation of mouse and EpiSC lines. We thank J.M. Brickman, W. Hamilton and J. Zylicz for critical reading of the manuscript and extensive discussion. This study is supported by the Novo Nordisk Foundation grant number NNF17CC0027852, and the Danish Council for Independent Research project grant 4183-00516A. Work supported by Marie Curie Reintegration Fellowship (H2020-MSCA-IF-2014_RI) to E.F.; and Lundbeckfonden PhD Fellowship (grant number R180-2014-2980) to X.G.

## Author contributions

L.M., X.G., N.A.M., A.M.D. and E.F. conceived, performed and analysed the experiments. X.G., N.A.M., L.M. and Q.W. performed the bioinformatics data analyses. X.G. and L.M. designed the CRISPR/Cas9 strategies. A.M.D. performed the lentiviral experiments. S.D.Ç. generated the *Pbx1^{fl/fl};Pbx2^{−/−};Cre^{pos}* EpiSC cell lines. L.M., X.G., N.A.M. and E.F. prepared the figures, wrote and edited the manuscript.

## Competing interests

The authors declare no competing interests.
