## [Peer Review File · Nature Communications]

REVIEWER COMMENTS

Reviewer #1 (Remarks to the Author):

This manuscript investigates the phenotype of a compound mutation in Pbx1/Pbx2 in mouse, where cells largely remain in a mesodermal progenitor state and fail to complete differentiation towards somites, meaning Pbx factors are part of the molecular circuitry leading to paraxial mesoderm differentiation. An array of molecular investigations using in vitro differentiation of EpiSCs shows that these factors increase chromatin accessibility at paraxial mesoderm effector genes, and the Wnt target Lef1 co-binds many of these regulatory regions. They focus on one key paraxial mesoderm transcriptional effector, *Msgn1*, showing that binding of Hox/Pbx is required for accessibility of the *Msgn1* promoter to Lef1, and to some extent, also *Tbra*.

This work is remarkable in taking a mutant phenotype from embryological studies through an in vitro cell differentiation model to global transcriptomic and epigenetic analyses and then focussed biochemical studies to show the regulatory logic of a key embryological process. Few other studies contain the multiplicity of approaches leading to a well-supported regulatory model. One of the key novelties of this work is that it demonstrates pioneer activity of Pbx proteins via targeted Crispr mutation of *Msgn1*, showing that this abolishes Lef1 binding. The data is overall well presented and the manuscript is well written.

I have a few points that I think the authors should address.

Major points

This study goes much further than others in the field to demonstrate the regulatory logic of cell differentiation in vivo. Nevertheless, the abstract and title lead us to believe that the study demonstrates a mechanism by which neuromesodermal progenitors choose between self-renewal and differentiation towards mesodermal fates. I believe this is not so clear from the present manuscript. Pbx proteins are near-ubiquitously expressed and so cannot themselves explain how a fate choice is made. It is clear from the present study that they allow response of cells to the mesoderm inducing signal Wnt. But what is not so clear is how, and whether, they act in NMPs to direct their choice between self-renewal and mesoderm fate (and therefore indirectly also neural fate). It seems to me that this is as big a mystery at the end of the study as at the beginning. This does not necessarily detract from the impact of the study, but might suggest a change in focus of the writing, or further demonstration of the role of Pbx proteins in NMPs. I have several linked questions here.

1. Is it clear that there are more NMPs in the mutants, at either stage analysed? This should appear in the single cell RNA-seq study, yet only the numbers of anterior or posterior paraxial mesoderm cells are compared. If not, how does this agree with the data in Fig 2, where there do seem to be more NMPs?

2. If there are not more NMPs in the single cell analysis (presumably the authors would have shown this if it were true?) then what is the identity of the cells expressing Sox2 and *Tbra* in Fig 2? Are they mesodermal progenitors that have only partially lost their NMP identity? The existence of a branch on the progenitor-to mesoderm trajectory containing only mutant cells both in vivo and in vitro suggests this may be the case: cells are not just paused on a normal differentiation trajectory but in an aberrant cell state. Further investigation here might highlight not just a blockade of differentiation, as has been demonstrated in the regulation of *Msgn1* by Pbx, but something more nuanced, and might give some insight into whether indeed Pbx proteins are acting in NMPs to control self renewal/differentiation, or act in NMP-derived mesoderm to consolidate onward mesoderm differentiation.

3. It's argued that the *Msgn1* regulatory mutation does not completely recapitulate the Pbx DKO phenotype and therefore that Pbx proteins have a dual role in NMPs and mesoderm differentiation (line 292-294). However this is not clearly shown by the data. It looks like some of the NMP markers are elevated in both cases (Fig 7c- *Nkx1-2*, *Sox2*).

It is interesting that anterior Hox family members are positively regulated by Pbx proteins, while posterior Hox members are negatively regulated at 24 h of EpiSC differentiation (Fig 4b,c). This suggests that Pbx proteins might have differential effects in different parts of the anteroposterior axis. The manuscript only examines the role of Hox-1 proteins, but is there any evidence from the embryo phenotype that Pbx proteins have different effects on the anterior versus posterior part of the axis? The statement in the discussion (line 326) that Pbx proteins may be balancing residence time in the progenitor region partially gets at this point, but is there any evidence from the aberrant mesoderm progenitors or the NMPs themselves that Hox expression is altered? Do the mutant cells have a Hox code that resembles a different time in development?

Minor points

Fig 5e- it is hard to see how Aldh1a2 fits the general point being made about Lef1-bound regions being differentially sensitive only in mesodermal genes, perhaps because it is shown at too low a resolution.

The paragraph about Hox codes (line 354-360) is quite unclear. I understand the 'hox code' to be the anteroposterior address of a part of the axis, programmed by the specific set of Hox genes expressed. So how is this 'not present in the N1 enhancer'?

Reviewer #2 (Remarks to the Author):

In this manuscript, Mariani et al. investigate the role of PBX and their cofactors, including HOX factors, in pre-somitic mesoderm differentiation using in vivo mouse models and a novel in vitro differentiation system.

5 main findings are presented:

- i. In Pbx1/2 double mutant embryos, neuromesodermal progenitors have a reduced ability to generate paraxial mesoderm and accumulate in the tailbud in a relatively undifferentiated state.
- ii. In an in vitro differentiation system in which EpiSCs are differentiated towards PSM, Pbx mutant cells exhibit a reduced capacity to acquire PSM identity and instead generate an increased number of NMPs/MPCs compared to WT cell cultures.
- iii. PBX proteins occupy regulatory regions associated with PSM genes, and modulate chromatin accessibility at these regions. Many of these regions also become occupied by LEF1 during PSM differentiation, and PBX factors are required for proper LEF1 occupancy at these regulatory sites.
- iv. Distinct PBX/TALE complexes are sequentially recruited at the *Msgn1* promoter during differentiation toward PSM, altering the chromatin landscape and activating transcription of *Msgn1*. A canonical PBX/HOX binding site is identified in the PBX-bound region of the promoter and can be bound by a PBX-MEIS2-HOXA1 complex in vitro.
- v. Mutation of the PBX/HOX site of the *Msgn1* promoter in cell culture leads to a loss/reduction of PBX1 and LEF1 binding at the promoter, loss/reduction of *Msgn1* expression, reduction of PSM and expansion of NMPs. This indicates that PBX-binding is required at this site to enable LEF1 binding and the subsequent transcription of *Msgn1* in response to WNT.

These findings lead the authors to propose a model in which PBX/MEIS/HOX1 complexes promote the WNT-mediated transcriptional response in paraxial mesoderm, by binding in cooperation with LEF1 to regulatory regions of key PM genes and promoting their expression.

This work makes several novel and very important findings regarding the roles of PBX factors and their co-factors in controlling the regulatory interactions that activate paraxial mesoderm development in mouse. The use of the EpiSC PSM differentiation protocol enables an incredibly detailed examination of the temporal changes in PBX and LEF1 binding and chromatin accessibility at key genes in the GRN. Furthermore, the detailed functional characterization and perturbation of this binding at the *Msgn1* promoter makes a compelling case for the role of PBX/TALE factors in enabling the transcriptional response of this gene to WNT signaling during PSM differentiation. However, I have a few points that I think should be addressed before I can fully recommend this manuscript to be published.

Major points:

- i. Description of uniformity/heterogeneity of cell states in the EpiSC to PSM differentiation protocol. Many of the claims in this work hinge on the validity of the cell culture system as a proxy for the differentiation of NMPs to PSM in the mouse tailbud. The use of scRNA-seq and analysis of marker gene co-expression in Fig3 provides convincing evidence that some of the cells in this culture system are behaving as expected. It would also be useful to have more description of the degree of uniformity/heterogeneity of gene expression and cell culture states in this system. What fraction of cells are responding appropriately to the differentiation protocol? Could there be a subpopulation of cells that are resistant to the differentiation treatments or have a delayed response? For instance, in Fig3i the histograms indicate that approx. 55% of cells at 48hpf are PSM, 5% are NMPs and 1% are MPCs. What are the rest of the cells that are not expressing these markers? If these contribute 'noise' to the system it would be useful to acknowledge and discuss this.
- ii. More evidence is required to claim direct involvement of HOX1 factors. A key claim in this work is that PBX/HOX1 complexes alter the NMP response to WNT signaling and drive differentiation to PSM. The authors make a convincing case for the direct role of PBX factors in this process. However, the involvement or requirement for HOX1 is inferred, but not demonstrated *in vivo*. The RNAseq data show that *HoxPG1* factors are expressed at the appropriate time, while the EMSAs nicely show that *HoxA1* can bind to the *Msgn1* regulatory element in a complex with PBX and MEIS *in vitro*. This suggests that HOX1 factors are strong candidates for regulating PSM differentiation with PBX. However, no data are presented that conclusively show that HOX1 factors bind to the *Msgn1* element, or other relevant PBX-bound elements, in NMPs *in vivo*. Thus, further evidence is required to make this claim. One possibility is Chip-qPCR to show HOX1 binding to the *Msgn1* element, either in the cell culture system or in embryonic tailbuds. A similar approach has been used to demonstrate HOXA1 binding to the *Raldh2* E3 enhancer in E8.5 embryos (Vitobello et al. 2011, *Dev Cell*), so it seems feasible here, unless appropriate antibodies are not available. If such demonstration of HOX1 binding *in vivo* is not technically feasible in this instance, then the claims regarding a direct role of HOX1 factors need to be toned down.
- iii. Potential role for NFY as a TALE collaborator at PBX-bound regions. In Fig4g the authors present NFY motifs amongst the enriched motifs in PBX-bound regions at various time-points during EpiSC to PSM differentiation. TALE factors and NFY have been shown to bind to many adjacent sites during early zebrafish development and to potentially form complexes (Ladam et al, 2018, eLIFE). This raises the prospect that NFY may also interact with PBX and other TALE factors and play a role during PSM differentiation in mouse, possibly as a pioneer factor. This deserves mention and discussion, especially since the work of Ladam et al describes how TALE factors employ distinct DNA motifs and protein partners (including Hox) at different embryonic stages in zebrafish, which seems to echo some of the interesting findings in this manuscript regarding dynamic combinatorial binding properties of TALE factors during mesoderm differentiation (e.g. Fig6 e,f).

Minor points:

- i. Line 215 - 'over 60% of the LEF1 targets were co-bound by PBX' - some info here about the

proximity of PBX and LEF1 peaks/sites at these targets would be useful. This could provide clues as to the nature of the interaction between TALE and LEF at regulatory sites.

ii. Fig4g. Presumably these are a selection of the enriched motifs at each timepoint. If so, please describe on what basis these were selected in the legend – where these the most enriched, or the most relevant/interesting-looking? It would be nice if all/more of the enriched motifs could be provided as supplementary data.

Best,
Hugo Parker

Reviewer #3 (Remarks to the Author):

Mariani et al, study the role of TALE/HOX complex in paraxial mesoderm specification from neuromesodermal progenitors (NMPs). While PBX/HOX complexes have been well described their role in driving paraxial mesoderm differentiation remains obscure. The study uses an elegant approach combining in vitro differentiation experiments and in vivo analysis of mouse double mutant embryos to show that TALE/HOX complexes control the gene regulatory networks that alter the NMP response to WNT signaling and activate the paraxial mesoderm program. Detailed analysis of Pbx1/2 double mutant embryos revealed that NMP cells remained trapped in an NMP/MPC state largely resembling the MSGN1 mutant phenotype. They further use CRISPR mutagenesis to introduce point mutations in the endogenous MSGN1 promoter, which resulted in reduced chromatin accessibility and LEF1 recruitment. These results demonstrated that PBX/HOX directly modulate chromatin accessibility and context dependent Wnt response. Overall, their study brings a comprehensive understanding of the process that regulates the differentiation of paraxial mesoderm from NMP cells and defines a novel role for TALE/HOX that act as a molecular switch to trigger alternative cellular responses to WNT signalling.

Minor points

In Figure 1C: The authors should make clear that MPCs co-express Brachyury, Msgn1 and TBX6 and then downregulate brachyury to adopt a PSM state (Msgn1+/Tbx6+). Also, in the dot plot Fig 1g the authors should add the expression of Msgn1.

Figure 3C: From the heatmap it is not clear if there are still NMP cells present at 36h and 48h of differentiation. It seems that the number of NMP cells using the described protocol is very low at 48h in wild type cells (less than 5% in Fig 3i). The authors should look more carefully if the cells that express posterior Hox genes are NMPs at 36 and 48 hours or pre neural tube cells that co-express Sox2/Nkx1.2.

We thank the referees for their positive assessment of our paper as well as their comments and suggestions which have helped us to improve our manuscript. In light of these comments, we have introduced some changes that include additional experiments and editing of the manuscript. Together with the original findings, they provide novel insights about WNT signalling response and important conceptual advances in the field of stem cell and developmental biology.

*“This work is remarkable in taking a mutant phenotype from embryological studies through an in vitro cell differentiation model to global transcriptomic and epigenetic analyses and then focussed biochemical studies to show the regulatory logic of a key embryological process. Few other studies contain the multiplicity of approaches leading to a well-supported regulatory model. One of the key novelties of this work is that it demonstrates pioneer activity of Pbx proteins via targeted Crispr mutation of *Msgn1*, showing that this abolishes *Lef1* binding. The data is overall well presented and the manuscript is well written.”* Reviewer #1

*“This work makes several novel and very important findings regarding the roles of PBX factors and their co-factors in controlling the regulatory interactions that activate paraxial mesoderm development in mouse. The use of the EpiSC PSM differentiation protocol enables an incredibly detailed examination of the temporal changes in PBX and LEF1 binding and chromatin accessibility at key genes in the GRN. Furthermore, the detailed functional characterization and perturbation of this binding at the *Msgn1* promoter makes a compelling case for the role of PBX/TALE factors in enabling the transcriptional response of this gene to WNT signaling during PSM differentiation.”* Reviewer #2

“The study uses an elegant approach combining in vitro differentiation experiments and in vivo analysis of mouse double mutant embryos to show that TALE/HOX complexes control the gene regulatory networks that alter the NMP response to WNT signaling and activate the paraxial mesoderm program [...] Overall, their study brings a comprehensive understanding of the process that regulates the differentiation of paraxial mesoderm from NMP cells and defines a novel role for TALE/HOX that act as a molecular switch to trigger alternative cellular responses to WNT signalling.” Reviewer #3

In our revised manuscript, we have addressed all the questions and suggestions made by the reviewers by performing new experiments, as well as rewriting the manuscript to explain and discuss more clearly the data and novel insights that our paper provides. In the manuscript, all the changes introduced in the text have been highlighted in red.

We have completely revised the abstract and introduced new or modified panels in the Figures as follows: Panels c and g in Fig. 1, Panels g and h in Supplementary Fig. 1, Panel c in Supplementary Fig. 2, Panel a in Supplementary Fig. 3, Panel c in Supplementary Fig. 5, Panel a in Supplementary Fig. 7. We added two new tables (Supplementary Table 1 and Supplementary Table 5) and two new references: Ladam *et al.*, 2018, *eLife* and Merabet *et al.*, 2011, *PLoS Genet.* All the changes are highlighted in red. However, changes in the figure order that are merely due to the addition of new panels have not been highlighted.

A point-by-point response to the reviewers' comments is provided below.

Referees' comments:

Referee #1 (Remarks to the Author):

This manuscript investigates the phenotype of a compound mutation in Pbx1/Pbx2 in mouse, where cells largely remain in a mesodermal progenitor state and fail to complete differentiation towards somites, meaning Pbx factors are part of the molecular circuitry leading to paraxial mesoderm differentiation. An array of molecular investigations using in vitro differentiation of EpiSCs shows that these factors increase chromatin accessibility at paraxial mesoderm effector genes, and the Wnt target Lef1 co-binds many of these regulatory regions. They focus on one key paraxial mesoderm transcriptional effector, Msgn1, showing that binding of Hox/Pbx is required for accessibility of the Msgn1 promoter to Lef1, and to some extent, also Tbra.

This work is remarkable in taking a mutant phenotype from embryological studies through an in vitro cell differentiation model to global transcriptomic and epigenetic analyses and then focussed biochemical studies to show the regulatory logic of a key embryological process. Few other studies contain the multiplicity of approaches leading to a well-supported regulatory model. One of the key novelties of this work is that it demonstrates pioneer activity of Pbx proteins via targeted Crispr mutation of Msgn1, showing that this abolishes Lef1 binding. The data is overall well presented and the manuscript is well written.

We thank the referee for expressing their enthusiasm for this work, and their appreciation of the diverse methods used to provide a broad understanding of the connection between PBX/HOX and WNT signalling during paraxial mesoderm differentiation.

This study goes much further than others in the field to demonstrate the regulatory logic of cell differentiation in vivo. Nevertheless, the abstract and title lead us to believe that the study demonstrates a mechanism by which neuromesodermal progenitors choose between self-renewal and differentiation towards mesodermal fates. I believe this is not so clear from the present manuscript.

We thank the reviewer for their comments and acknowledge their criticisms. We have rephrased the abstract to make it clearer as suggested by the referee. However, we retained the title, given it does not refer to the self-renewal of neuromesodermal progenitors (NMPs).

In this manuscript, we dissected the mechanism by which NMPs interpret WNT signalling leading to a shift in the balance from NMP expansion towards pre-somitic mesoderm (PSM) differentiation. We found that induction of the PSM program specifically relies on the synergistic activity of the PBX transcription factors and the WNT-effector LEF1. We identified a WNT-HOX code that unlocks paraxial mesodermal genes, making them accessible to LEF1 and authorizing WNT signalling response towards paraxial mesoderm. Activation of the WNT-HOX code makes NMPs acquire a mesodermal fate, leave the progenitor zone (NMP expansion zone) and progress through the PSM. In Pbx mutant embryos, cells adopting paraxial mesodermal fate are trapped in a transition state, co-expressing signature markers of both PSM and NMPs, and accumulate in the tailbud.

The comment of the reviewer made us reconsider the role played by PBX in controlling the balance between self-renewal and differentiation, and we thank him/her for raising this point. In the manuscript, we demonstrate that paraxial mesoderm induction relies on TALE/HOX combinatorial activity that simultaneously represses NMP genes and activates the paraxial mesoderm differentiation program. Our data suggest that PBX negatively control genes that are

important for keeping the NMP state, like *Sox2*, *T-Bra*, *Cdx2*, although their role in NMP self-renewal is less clear. For this reason, we modified the panel in Supplementary Fig. 2c, removing the link between *Sox2*, *T-Bra*, *Cdx2* and self-renewal and stating instead the importance of such genes for maintaining the NMP state. We agree with the reviewer that we do not have strong evidences that PBX control the self-renewal of NMPs. For this reason, we decided to tone down the claims about the function of PBX in NMP self-renewal, mentioning only a potential role of TALE for maintaining the NMP state.

Pbx proteins are near-ubiquitously expressed and so cannot themselves explain how a fate choice is made. It is clear from the present study that they allow response of cells to the mesoderm inducing signal *Wnt*. But what is not so clear is how, and whether, they act in NMPs to direct their choice between self-renewal and mesoderm fate (and therefore indirectly also neural fate). It seems to me that this is as big a mystery at the end of the study as at the beginning. This does not necessarily detract from the impact of the study, but might suggest a change in focus of the writing, or further demonstration of the role of *Pbx* proteins in NMPs. The PBX proteins, PBX1 and PBX2, are indeed ubiquitously expressed in the mouse tailbud, as shown by immunofluorescence (IF) studies in Supplementary Fig. 1a. However, the PBX proteins work through a combinatorial logic (Merabet *et al.*, 2011, *PLoS Genet*). PBX specific activity in PSM differentiation is dictated by the formation of a multimeric complex with MEIS2 and HOX-1, which are upregulated in the progenitors that adopt the paraxial mesodermal fate (Supplementary Fig. 7a). The unique combinatorial interaction of PBX/MEIS2/HOX-1 proteins generates a DNA-binding complex able to recognize specific DNA-binding motifs on the paraxial mesodermal genes. Furthermore, the PBX/MEIS2/HOX-1 complex increases chromatin accessibility enabling the WNT-effector LEF1 to bind to regulatory regions of PSM genes, leading to their subsequent activation.

We thank the reviewer for raising this very important question making it apparent that our original manuscript was unclear. In the revised manuscript, we further clarify the combinatorial logic of the PBX proteins.

As for the role of PBX in the neural fate, we did not investigate the function of PBX proteins in the NMPs fated to the neural lineages. We have not tested if the “trapped” *Pbx-DKO* NMPs/MPCs have neural potential, which lies beyond the scope of the current study. However, we can report that we did not observe evident neural defects in the *Pbx* mutant embryos. We did not detect the formation of ectopic neural tubes, as reported for *Wnt3a* and *Tbx6* mutants, thereby making the phenotype of *Pbx* embryos more similar to the one exhibited by the *Msgn1* mutants.

I have a few points that I think the authors should address.

Linked questions:

1. Is it clear that there are more NMPs in the mutants, at either stage analysed? This should appear in the single cell RNA-seq study, yet only the numbers of anterior or posterior paraxial mesoderm cells are compared. If not, how does this agree with the data in Fig 2, where there do seem to be more NMPs?

We thank the reviewer for their comments and for pointing out the lack of clarity in Fig. 1 and Fig. 2. In Figure 1, we analysed the tailbud of WT and mutant embryos by single-cell RNA-sequencing (scRNA-seq). The different cell populations were clustered accordingly to their global transcriptome. This allowed the identification of three main clusters: NMPs, mesodermal progenitor cells/posterior PSM (MPCs/pPSM) and anterior PSM (aPSM). While we could not

find significant differences in term of cell numbers in the NMP cluster between WT and *Pbx* mutants, when we considered the expression of selected lineage genes, we noticed that *Pbx-DKO* NMPs carry a mixed signature, expressing also some paraxial mesoderm genes, as described in the new panel in Supplementary Figure 1g. Thus, it is possible that some of the mutant NMPs clustered as MPCs.

Furthermore, it is technically very challenging to capture the NMPs *in vivo* by scRNA-seq because they are few (in particular at E9.5) and because the key lineage marker *Sox2* used for their identification is expressed at low levels (Edri *et al.*, 2019, *Development*; Dias *et al.*, 2020, *eLife*), and can therefore be below the threshold of detection. All these aspects make it quite challenging to characterize NMPs *in vivo*. In our *in vitro* system, the number of NMPs is higher and thus it is easier to profile them by scRNA-seq (Fig. 3b,c,f,g).

In Figure 2, instead of considering the global transcriptome, we attributed cell identity using only three signature markers/antibodies. Cells that are T-BRA^{pos};SOX2^{pos} were considered NMPs, while cells expressing both TBX6 and T-BRA were counted as MPCs. The IF allowed us to overcome the experimental constraints in detecting *Sox2* transcripts by scRNA-seq. The specificity and sensitivity of the SOX2 antibody allowed a better detection of the cells expressing SOX2, making the analysis and identification of NMPs by IF more robust.

In conclusion, we demonstrated that in *Pbx* mutants, NMPs exhibit reduced ability to generate PSM, are trapped in a NMP/MPC transition state and accumulate in the tailbud. In agreement with the reviewer's comment, in the *Pbx* mutants the trajectory that goes from NMPs to paraxial mesoderm is compromised and mutant cells could be blocked at different stages along differentiation (Fig. 1h). However, we cannot discriminate clearly these intermediate states due to the current absence of specific signature markers.

*2. If there are not more NMPs in the single cell analysis (presumably the authors would have shown this if it were true?) then what is the identity of the cells expressing Sox2 and Tbra in Fig 2? Are they mesodermal progenitors that have only partially lost their NMP identity? The existence of a branch on the progenitor-to mesoderm trajectory containing only mutant cells both in vivo and in vitro suggests this may be the case: cells are not just paused on a normal differentiation trajectory but in an aberrant cell state. Further investigation here might highlight not just a blockade of differentiation, as has been demonstrated in the regulation of *Msgn1* by *Pbx*, but something more nuanced, and might give some insight into whether indeed *Pbx* proteins are acting in NMPs to control self-renewal/differentiation, or act in NMP-derived mesoderm to consolidate onward mesoderm differentiation.*

The rationale of Fig. 2 was to provide a spatial localization of the trapped NMP/MPCs, as we analysed the distribution of SOX2^{pos}/T-BRA^{pos} (NMPs) and T-BRA^{pos}/TBX6^{pos} (MPCs) cells in WT and *Pbx* mutant embryos. We specifically addressed the distribution of progenitor populations in the caudal and medial parts of the CLE, where cells preferentially adopt a PSM fate (Fig. 2b). In other words, we concentrated our analysis on cells that are fated to adopt a paraxial mesodermal fate. We fully agree with the reviewer's comment about the possible generation of an aberrant cell state in the *Pbx* mutants, and indeed, we addressed this aspect in the revised manuscript. However, it is not possible to discriminate unequivocally these intermediate/aberrant states with these three markers.

In this manuscript, we specifically concentrated on the transition of NMPs towards MPCs, where we demonstrated a role for PBX in consolidating paraxial mesoderm differentiation. However, our data suggest that PBX complexes may directly regulate genes important for the maintenance

of the NMP state, like *Sox2*, *T-Bra* and *Cdx2*. It will be interesting to mutagenize the TALE-WNT module in the regulatory regions of these genes and test the effect on the NMP state and self-renewal properties. This approach might clarify if and how the TALE-WNT code has a role in the maintenance of NMPs, and it could be an interesting subject for future follow-up investigations.

3. *It's argued that the *Msgn1* regulatory mutation does not completely recapitulate the *Pbx* DKO phenotype and therefore that *Pbx* proteins have a dual role in NMPs and mesoderm differentiation (line 292-294). However this is not clearly shown by the data. It looks like some of the NMP markers are elevated in both cases (Fig 7c- *Nkx1-2*, *Sox2*).*

We thank the reviewer for their thoughtful comment and apologize for not having properly described this important point. After a deeper examination of the *pMsgn1-mut* phenotype, we have revised our interpretation of the results. Indeed, the global loss of mesodermal markers and the enhanced expression of NMP genes in both *Pbx-DKO* and *pMsgn1-mut* lines as compared to wild-type (WT) indicate their shared inability to progress towards the paraxial mesodermal fate, confirming the role of *Pbx* and *Msgn1* in consolidating PSM differentiation. However, the distinct profile of *Pbx-DKO* and *pMsgn1-mut* cells suggests that these mutations may affect PSM differentiation with a different degree. In the *Pbx* mutant lines, the higher levels of *T-Bra*, *Fgf8*, *Cdx2* and *Wnt3a*, and the concomitant low levels of *Msgn1*, *Cdh2*, *Hes7*, *Dll1*, *Snai1* and *Tbx6*, support an aberrant NMP/MPC state. The *Pbx* mutant phenotype suggests multiple scenarios, including compensatory mechanisms by other TALEs, or alternative effects due to PBX fulfilling multiple roles on specific targets that are different from the genes controlled uniquely by MSGN1. In *pMsgn1-mut* lines, the phenotype is apparently exacerbated, as shown by increased expression of *Cdh1*, *Sox2* and more specifically by the absence of *T-Bra* and other mesodermal genes (*Tbx6*), including epithelial-to-mesenchymal transition (EMT) markers (*Cdh2*, *Snai1*), resulting in a complete loss of paraxial mesoderm formation. Accordingly, we modified the description of the cell lines in the revised manuscript.

It is interesting that anterior Hox family members are positively regulated by Pbx proteins, while posterior Hox members are negatively regulated at 24 h of EpiSC differentiation (Fig 4b,c). This suggests that Pbx proteins might have differential effects in different parts of the anteroposterior axis. The manuscript only examines the role of Hox-1 proteins, but is there any evidence from the embryo phenotype that Pbx proteins have different effects on the anterior versus posterior part of the axis? The statement in the discussion (line 326) that Pbx proteins may be balancing residence time in the progenitor region partially gets at this point, but is there any evidence from the aberrant mesoderm progenitors or the NMPs themselves that Hox expression is altered? Do the mutant cells have a Hox code that resembles a different time in development?

We thank the referee for raising the question about the patterning defects in *Pbx* mutants and contribution of the HOX code to axial elongation that were only partially addressed in the original manuscript. Anterior and posterior *Hox* genes have opposing roles in initiating and terminating axial elongation. The most posterior *Hox* genes (*Hox10-13*) reduce mesoderm ingression by repressing WNT and are responsible for body axis termination (Young *et al.*, 2009, *Dev Cell* and Denans *et al.*, 2015, *eLife*). Conversely, the anterior *Hox* genes (*Hox1-8*) control paraxial mesoderm formation by modulating cell ingression into the primitive streak (Iimura *et al.*, 2006, *Nature*). Nevertheless, the mechanism by which anterior *Hox* genes promote paraxial

mesoderm specification remains largely unknown and it does represent the aspect addressed in our manuscript.

We examined only HOX group1 because they are specifically enriched in the NMPs and early MPCs (Supplementary Fig. 1e and Fig. 3c) and therefore HOX-1 represent the best candidate to address the role of the HOX code during the NMP-to-MPC transition.

Regarding the anterior/posterior axis defects of the *Pbx* mutants, we observed that the anterior somites, from 1 to 7, are not significantly affected, as shown by gross morphology in Fig. 1a and Supplementary Fig. 1b. Similarly, mouse embryos carrying mutations in *Msgn1*, one of the major targets of PBX, have been reported to exhibit aberrant morphogenesis of somites posterior to 6/7, while the anterior somites 1/6 are mildly affected (Yoon *et al.*, 2000, *Genes Dev*). *Msgn1*, in addition to controlling paraxial mesoderm differentiation, promotes the migration of MPCs from the progenitor zone to the PSM territory, ultimately regulating the flux of cells acquiring PSM fate. *Pbx-DKO* MPCs exhibit reduced migratory ability as visualized along PSM *in vitro* differentiation in time-lapse videos (Supplementary Videos 1,2 and Supplementary Fig. 6b). As a result, the flux of the migrating MPCs in the *Pbx* mutant is reduced and they accumulate in the progenitor zone. Indeed, *Pbx* mutants, like *Msgn1* mutants, possess an enlarged tailbud, presumably resulting from the accumulation of mesoderm cells that fail to migrate.

Previous studies have shown that the HOX code is critical for axial elongation because it provides the positional information to the axial progenitors. *Hox* genes are collinearly expressed during axial elongation, establishing a spatio-temporal gradient along the anterior-posterior axis. This HOX clock provides NMPs with spatial coordinates and induces MPCs to progressively acquire a more posterior identity, ensuring that cell fate and position are intrinsically linked. It has been shown that early MPCs and NMPs expressed more 3' *Hox* and thus have an anterior identity, while progenitors arising later in time expressed more 5' *Hox* genes, possess a more posterior identity and migrate later to the PSM (Gouti *et al.*, 2017, *Dev Cell*). In our systems, while *in vivo* *Hox-1* genes have comparable expression in control and *Pbx* mutants, their levels are reduced in the *Pbx-DKO* cells *in vitro*. This effect could be due to compensatory mechanisms occurring in the embryo that are not present in our culture conditions. Although these are interesting aspects to investigate, due to the space constraints of the journal we will reserve further discussion and analyses to future manuscripts.

Nonetheless, as noticed by the reviewer, and shown in the attached Fig. 1 and Supplementary Fig. 1h, we found that the NMPs and MPCs in the *Pbx* mutants show higher expression of the 5' *Hox* genes at E8.5 and E9.5, thus bearing a more posterior identity in comparison to control. It is difficult to establish whether in the *Pbx* mutants the posterior HOX code is already present in the nascent NMPs or it is a feature acquired by the accumulating NMPs in the tailbud.

As suggested by the reviewer, it is also possible that the mutant NMP/MPCs bearing the 5' *Hox* signature could provide anterior paraxial mesoderm structures with more posterior traits. This hypothesis is in agreement with the rostral shift of the axial *Hox* gene expression and the hindlimb positioning observed in the *Pbx-com* mutants (Capellini *et al.*, 2008, *Dev Biol*).

This is an interesting interpretation of the *Pbx* mutant phenotype and has been discussed in the revised manuscript. We thank the reviewer for raising this important observation that has helped us improve the interpretation of our results.

Fig. 1

Fig. 1. The *Pbx* mutant NMPs/MPCs show higher expression of 5' *Hox* genes. a-b, Expression analyses of *Hox* genes within the NMPs (a) and the MPCs/pPSM (b) clusters isolated from control and *Pbx* mutant tailbuds at embryonic days (E) 8.5 and 9.0. Colour bars indicate the intensity associated with normalised expression values. *Pbx-com*: *Pbx1*^{-/-};*Pbx2*^{+/-} compound mutant; *Pbx-DKO*: *Pbx1*^{-/-};*Pbx2*^{-/-} double-knockout mutant.

Minor points

Fig 5e- it is hard to see how *Aldh1a2* fits the general point being made about *Lef1*-bound regions being differentially sensitive only in mesodermal genes, perhaps because it is shown at too low a resolution.

We fully agree with this comment about the low resolution of the PBX/HOX and LEF1 binding peaks in the *Aldh1a2* regulatory region. We substituted the *Aldh1a2* panel with a higher magnification where the reduced chromatin accessibility in the *Pbx* mutant cells is clearly visible.

The paragraph about *Hox* codes (line 354-360) is quite unclear. I understand the 'hox code' to be the anteroposterior address of a part of the axis, programmed by the specific set of *Hox* genes expressed. So how is this 'not present in the N1 enhancer'?

We apologize for being unclear. In the paragraph 354-360, we addressed the evolutionary importance of the PBX/HOX-WNT module. While in eutherians, axial elongation relies on a pool of expanding NMPs, in lower vertebrates like zebrafish the PSM maturation is supported by depleting progenitors (NMPs do not expand in zebrafish) (Steventon and Martinez Arias, 2017 *Dev Biol*). In higher vertebrates, a network of TFs that includes SOX2 and specifically the N1 responsive element, are associated with the NMP state and self-renewal properties (Takemoto *et al.*, 2011, *Nature*). We observed that the PBX/HOX-WNT module is conserved within eutherians, as highlighted by the evolutionary tracks of Fig. 5e, which includes the N1 element in the *Sox2* regulatory sequence. The N1 responsive element and the PBX/HOX-WNT module are not present in the *Sox2* regulatory region of lower vertebrates. These intriguing observations have led us to speculate that during vertebrate evolution, WNT and HOX codes may have been

co-opted to balance expansion and differentiation of paraxial mesoderm progenitors for axial elongation.

In the revised manuscript, we have highlighted the N1 element in Fig. 5e and rephrased the sentence in the discussion.

Reviewer #2 (Remarks to the Author):

In this manuscript, Mariani et al. investigate the role of PBX and their cofactors, including HOX factors, in pre-somitic mesoderm differentiation using in vivo mouse models and a novel in vitro differentiation system.

5 main findings are presented:

- i. In Pbx1/2 double mutant embryos, neuromesodermal progenitors have a reduced ability to generate paraxial mesoderm and accumulate in the tailbud in a relatively undifferentiated state.*
- ii. In an in vitro differentiation system in which EpiSCs are differentiated towards PSM, Pbx mutant cells exhibit a reduced capacity to acquire PSM identity and instead generate an increased number of NMPs/MPCs compared to WT cell cultures.*
- iii. PBX proteins occupy regulatory regions associated with PSM genes, and modulate chromatin accessibility at these regions. Many of these regions also become occupied by LEF1 during PSM differentiation, and PBX factors are required for proper LEF1 occupancy at these regulatory sites.*
- iv. Distinct PBX/TALE complexes are sequentially recruited at the Msgn1 promoter during differentiation toward PSM, altering the chromatin landscape and activating transcription of Msgn1. A canonical PBX/HOX binding site is identified in the PBX-bound region of the promoter and can be bound by a PBX-MEIS2-HOXA1 complex in vitro.*
- v. Mutation of the PBX/HOX site of the Msgn1 promoter in cell culture leads to a loss/reduction of PBX1 and LEF1 binding at the promoter, loss/reduction of Msgn1 expression, reduction of PSM and expansion of NMPs. This indicates that PBX-binding is required at this site to enable LEF1 binding and the subsequent transcription of Msgn1 in response to WNT.*

These findings lead the authors to propose a model in which PBX/MEIS/HOX1 complexes promote the WNT-mediated transcriptional response in paraxial mesoderm, by binding in cooperation with LEF1 to regulatory regions of key PM genes and promoting their expression.

This work makes several novel and very important findings regarding the roles of PBX factors and their co-factors in controlling the regulatory interactions that activate paraxial mesoderm development in mouse. The use of the EpiSC PSM differentiation protocol enables an incredibly detailed examination of the temporal changes in PBX and LEF1 binding and chromatin accessibility at key genes in the GRN. Furthermore, the detailed functional characterization and perturbation of this binding at the Msgn1 promoter makes a compelling case for the role of PBX/TALE factors in enabling the transcriptional response of this gene to WNT signaling during PSM differentiation. However, I have a few points that I think should be addressed before I can fully recommend this manuscript to be published.

We thank the referee for their positive assessment of our manuscript and for acknowledging the novelty and importance of the PBX/TALE-mediated regulatory network in axial elongation. We also thank the referee for the suggestions that have helped us improve our manuscript.

Major points:

i. Description of uniformity/heterogeneity of cell states in the EpiSC to PSM differentiation protocol. Many of the claims in this work hinge on the validity of the cell culture system as a proxy for the differentiation of NMPs to PSM in the mouse tailbud. The use of scRNA-seq and analysis of marker gene co-expression in Fig3 provides convincing evidence that some of the cells in this culture system are behaving as expected. It would also be useful to have more description of the degree of uniformity/heterogeneity of gene expression and cell culture states in this system. What fraction of cells are responding appropriately to the differentiation protocol? Could there be a subpopulation of cells that are resistant to the differentiation treatments or have a delayed response? For instance, in Fig3i the histograms indicate that approx. 55% of cells at 48hpf are PSM, 5% are NMPs and 1% are MPCs. What are the rest of the cells that are not expressing these markers? If these contribute 'noise' to the system it would be useful to acknowledge and discuss this.

We thank the reviewer for their comments and critical feedback. We evaluated the efficiency of differentiation by analysing the expression of the primitive streak marker T-BRA in the attached Fig. 2. Moreover, we added a UMAP panel in Supplementary Fig. 3a showing that all the cells at 12 h of differentiation were *T-Bra*^{pos}, indicating an efficient primitive streak induction. Furthermore, PSM cells at 48 h specifically express *Tbx6* and are negative for the NMP marker *Sox2*. Overall these results confirmed the efficiency of the differentiation system towards a mesodermal fate.

To address the remarks about the percentage of progenitors in Fig. 3 of the manuscript, it is helpful to begin with the rationale behind scoring: only cells that expressed high levels of T-BRA, SOX2, or TBX6 were counted. Therefore, cells expressing low levels of these markers were not included in the calculation. Although such an approach would neglect transient progenitor populations, we adopted this criterium to be 100% certain of only considering cells with the correct signature.

In the revised manuscript, we have now included a discussion of the counting method in the Methods section and a narrative about the criteria chosen for the counting.

Fig. 2

Fig. 2. Epiblast stem cells are efficiently differentiated *in vitro* to anterior primitive streak. **a**, Top: Schematics and scRNA-seq of epiblast stem cells (EpiSCs) differentiation towards anterior primitive streak (APS). Unsupervised clustering and UMAP visualisation reveal two clearly separated clusters, indicating efficient differentiation towards primitive streak. Bottom: UMAP of EpiSCs and APS clusters. Cells are coloured by expression of the EpiSC marker *Utf1* (left) and the PS marker *T-Bra* (right). Importantly, all cells express *T-Bra* after 12 h of differentiation. Colour bars indicate the intensity associated with normalised expression values. **b**, Representative immunofluorescence staining for the PS marker T-BRA (green) at 12 h of differentiation. Scale bar: 50 μm.

ii. More evidence is required to claim direct involvement of *HOX1* factors. A key claim in this work is that *PBX/HOX1* complexes alter the NMP response to WNT signaling and drive differentiation to PSM. The authors make a convincing case for the direct role of *PBX* factors in this process. However, the involvement or requirement for *HOX1* is inferred, but not demonstrated *in vivo*. The RNAseq data show that *HoxPG1* factors are expressed at the appropriate time, while the EMSAs nicely show that *HoxA1* can bind to the *Msgn1* regulatory element in a complex with *PBX* and *MEIS* *in vitro*. This suggests that *HOX1* factors are strong candidates for regulating PSM differentiation with *PBX*. However, no data are presented that conclusively show that *HOX1* factors bind to the *Msgn1* element, or other relevant *PBX*-bound elements, in NMPs *in vivo*. Thus, further evidence is required to make this claim. One possibility is Chip-qPCR to show *HOX1* binding to the *Msgn1* element, either in the cell culture system or in embryonic tailbuds. A similar approach has been used to demonstrate *HOXA1* binding to the *Raldh2* E3 enhancer in E8.5 embryos (Vitobello et al. 2011, Dev Cell), so it seems feasible here, unless appropriate antibodies are not available. If such demonstration of *HOX1* binding *in vivo* is not technically feasible in this instance, then the claims regarding a direct role of *HOX1* factors need to be toned down.

We tried to perform ChIP-qPCR using *in vitro* differentiated cells, as described in the attached Figure 3, by employing 5 different antibodies:

- *HOXA1* from R&D Systems (Cat. Nr.: AF5014);
- *HOXB1* from R&D Systems (Cat. Nr.: AF6318);

- HOXB1 from Sigma (Cat. Nr.: SAB1409201);
- HOXB1 from Invitrogen (Cat. Nr.: PA593101);
- HOXA1/B1/D1 from Invitrogen (Cat. Nr.: PA5103889).

Besides ChIP-qPCR, we also performed Cut & Run, followed by sequencing, using the abovementioned antibodies with tailbuds obtained from E8.5 mouse embryos, since it would have allowed us to overcome the cross-linking step that can promote epitope masking (attached Figure 3).

Sadly, none of these antibodies and approaches have proven to efficiently and specifically immunoprecipitate the HOX-1 complexes from any tested gene. Unfortunately, the HOXA1 antibody (N20 sc17146X) described in Vitobello *et al.*, 2011, *Dev Cell* paper is no longer in production, and in general, the antibodies against the HOX proteins are well-known for their reduced immunogenic output.

Although we understand that the definitive proof of HOX-1 recruitment on the *Msgn1* promoter will conclusively demonstrate the formation of a PBX/HOX complex, we would like to emphasize that we provided other evidences strongly supporting the formation of the HOX complex on the *Msgn1* regulatory region.

Therefore, before detailing changes to the narrative, it may be helpful to recapitulate the reasoning that led to the current model.

- PBX are well-established HOX cofactors (Mann & Lelli, 2009, *Curr Top Dev Biol*). Studies on direct interaction of HOX/PBX date back to the '90s with the seminal work of Wieschaus in *Drosophila* (Peifer and Wieschaus, 1990, *Genes Dev*).
- The PBX-bound motif in the *Msgn1* regulatory region corresponds to a PBX/HOX-1 motif (Mann & Lelli, 2009, *Curr Top Dev Biol*).
- We demonstrated by EMSA that PBX/MEIS2/HOX-1 complexes can be assembled on the oligonucleotide containing the *Pbx/Hox-1* motif designed on the *Msgn1* regulatory region. Furthermore, we showed that specific point mutations within the *Pbx/Hox-1* binding site abrogate the formation of the complex.
- Remarkably, in our *in vitro* differentiation system, expression and accessibility of the *Msgn1* regulatory region correlate with the expression of *Hox* group1 proteins.
- PBX/HOX-A1 complexes are detected on the *E3* regulatory element of *Aldh1a2* in E8.5 murine tailbuds as reported by Vitobello *et al.*, 2011, *Dev Cell* and pointed out by the reviewer. The newly identified PBX-LEF1 module of *Aldh1a2* corresponds to the *E3* enhancer characterized by Vitobello *et al.* Thus, at least for the *Aldh1a2* gene, the activity of the WNT-HOX integrated code supports *Aldh1a2* expression.

Ultimately, PBX proteins are obligated binding partners of the HOX-1 proteins. We do not have any good reasons for thinking that they will work differently in the case of *Msgn1*.

We understand that the ChIP-seq data for HOX-1 could provide the definitive proof of the PBX/HOX-1 complex formation *in vivo*, however these experiments are not feasible with the available resources. As suggested by the reviewer, we therefore toned down our claims about the PBX/HOX complex activity and we have substituted the term HOX complex with TALE complex.

Fig. 3

a

Antibody (clonality, host)	Company	Catalog nr.	Nr.
HOXA1 (polyclonal, Goat)	R&D Systems	AF5014	1
HOXB1 (polyclonal, Sheep)	R&D Systems	AF6318	2
HOXB1 (monoclonal, Mouse)	Sigma	SAB1409201	3
HOXB1 (polyclonal, Rabbit)	Invitrogen	PA593101	4
HOXA1/B1/D1 (polyclonal, Rabbit)	Invitrogen	PA5103889	5

b

c

Fig. 3. None of the available antibodies can efficiently immunoprecipitate HOX-1 complexes. **a**, Antibodies tested for ChIP and Cut & Run. **b**, ChIP-qPCR analyses of HOX-1 proteins on the *Hoxb1*, *Meis2* and *Msgn1* loci at indicated time-points of *in vitro* PSM differentiation. The *Hoxb1* enhancer region (top, described in Ferretti *et al.*, 2005, *Mol Cell Biol*) and the 3' UTR of *Meis2* (bottom, described in De Kumar *et al.*, 2017, *Genome Res*) were tested as positive controls for HOXB1 and HOXA1 binding, respectively. **c**, Cut & Run coverage tracks for the *Hoxb1* (top), *Aldh1a2* (bottom, described in Vitobello *et al.*, 2011, *Dev Cell*) and *Msgn1* loci. Threshold of vertical viewing range of data based on RPKM values is noted. Conservation across vertebrates is indicated in green.

iii. *Potential role for NFY as a TALE collaborator at PBX-bound regions. In Fig4g the authors present NFY motifs amongst the enriched motifs in PBX-bound regions at various time-points during EpiSC to PSM differentiation. TALE factors and NFY have been shown to bind to many adjacent sites during early zebrafish development and to potentially form complexes (Ladam et al, 2018, eLIFE). This raises the prospect that NFY may also interact with PBX and other TALE factors and play a role during PSM differentiation in mouse, possibly as a pioneer factor. This deserves mention and discussion, especially since the work of Ladam et al describes how TALE factors employ distinct DNA motifs and protein partners (including Hox) at different embryonic stages in zebrafish, which seems to echo some of the interesting findings in this manuscript regarding dynamic combinatorial binding properties of TALE factors during mesoderm differentiation (e.g. Fig6 e,f).*

Based on the remarks of the reviewer, we have revised the Discussion section. We added the NFY reference and emphasized the hypotheses that NFY could be involved in combinatorial binding with the TALE proteins (Ladam et al, 2018, eLife). We are grateful to the reviewer for pointing out this interesting paper.

Although we cannot exclude that combined PBX/NFY motifs may play a general role in promoting access to enhancers, the fact that these motifs have been found in several other contexts (Penkov et al., 2013, Cell Rep) and that NFY is broadly expressed make the claim of specificity for the mesodermal genes a bit weak. Although there are NFY (CCAAT) binding motifs in the *Msgn1* promoter, they are not in close proximity to the PBX-bound region, advocating against a direct interaction with the TALE proteins. However, without performing ChIP-seq and detailed point mutation analyses of the NFY-binding sites, it is difficult to assess the contribution of NFY in paraxial mesoderm differentiation. This aspect could be an interesting subject for future follow-up investigations.

We hope that, with all the changes listed above, the reviewer will find the manuscript more compelling.

Minor points

i. Line 215 – ‘over 60% of the LEF1 targets were co-bound by PBX’ – some info here about the proximity of PBX and LEF1 peaks/sites at these targets would be useful. This could provide clues as to the nature of the interaction between TALE and LEF at regulatory sites.

As suggested by the reviewer, to have a clearer picture of the LEF1 and TALE protein interactions, we included in the Supplementary Figure 5 of the revised manuscript a panel describing the proximity of PBX and LEF1 ChIP-seq peaks at 24 h of differentiation. We correlated the number of events where LEF1 and PBX binding sites get together, to their relative distance. We calculated the distance focusing on LEF1 peaks and looking for the closest PBX peaks or vice versa considering the PBX peaks and seeking for the closest LEF1 peaks. We observed that the distance between LEF1 and PBX-binding sites is flexible and varies at different regulatory regions. We found that 479 LEF1 peaks and 500 PBX peaks, corresponding to the 61% and 26% of the total peaks, respectively, partially overlap. Given that the average size of the LEF1 and PBX peaks is 292 bp and 526 bp, respectively, the overlapping PBX and LEF1 peaks are distributed in a variable interval of 409 bp. These distributions could indicate that the PBX and LEF1-binding sites could be located at a relatively close distance allowing the formation of multimeric complexes.

However, the variable distance between PBX and LEF1-binding sites could also support cooperative binding, without direct interactions, allowing the specific recruitment of other TFs that further stabilize the nucleosome-depleted regions triggered by the PBX complexes. Thus, changes in chromatin accessibility initiated by the PBX complexes, could mediate the recruitment of additional proteins, including ATP-dependent chromatin remodelling factors. In addition, a small proportion of the PBX and LEF1-binding events occurs at long distance. To this regard, LEF1 is an architectural protein, that mediates locus-specific DNA looping to facilitate the interaction of distant cis-regulatory elements and the recruitment of β -catenin. In this context, LEF1 might sustain long-distance interaction with PBX complexes. Alternatively, PBX and their cofactors could fine-tune the DNA bending or stabilize the looping. Considering all these possible scenarios, it is very challenging to obtain a definitive picture about the TALE and LEF1 interactions. Overall, the flexible arrangement of the distance between TALE and LEF1-binding sites supports the hypothesis of recruitment of additional TFs. The promiscuity of TF-binding site arrangements is the basic feature for TF cooperativity and is extensively used for fine-tuning gene regulation. In summary, in response to the reviewer's comment, we developed further the narrative surrounding the nature of interactions between the TALE and LEF1 proteins. However, given the space constraints of the journal, the results and further discussion were largely developed in the Supplementary Figure and Methods section.

ii. Fig4g. Presumably these are a selection of the enriched motifs at each timepoint. If so, please describe on what basis these were selected in the legend – where these the most enriched, or the most relevant/interesting-looking? It would be nice if all/more of the enriched motifs could be provided as supplementary data.

In Fig. 4, only the most relevant enriched motifs are listed. We excluded the motifs with low p-values. We agree with the reviewer that having the complete list of these motifs could be a relevant resource for the reader of *Nature Communications*. In the Supplementary Table 5, we listed all the enriched motifs with their correlated p-values.

Reviewer #3 (Remarks to the Author):

Mariani et al, study the role of TALE/HOX complex in paraxial mesoderm specification from neuromesodermal progenitors (NMPs). While PBX/HOX complexes have been well described their role in driving paraxial mesoderm differentiation remains obscure. The study uses an elegant approach combining in vitro differentiation experiments and in vivo analysis of mouse double mutant embryos to show that TALE/HOX complexes control the gene regulatory networks that alter the NMP response to WNT signaling and activate the paraxial mesoderm program. Detailed analysis of Pbx1/2 double mutant embryos revealed that NMP cells remained trapped in an NMP/MPC state largely resembling the MSGN1 mutant phenotype. They further use CRISPR mutagenesis to introduce point mutations in the endogenous MSGN1 promoter, which resulted in reduced chromatin accessibility and LEF1 recruitment. These results demonstrated that PBX/HOX directly modulate chromatin accessibility and context dependent Wnt response. Overall, their study brings a comprehensive understanding of the process that regulates the differentiation of paraxial mesoderm from NMP cells and defines a novel role for TALE/HOX that act as a molecular switch to trigger alternative cellular responses to WNT signalling.

We thank the referee for their positive comments on our manuscript. By adding all the additional requested analyses, we have now hopefully addressed all of the concerns.

Minor points

In Figure 1C: The authors should make clear that MPCs co-express Brachyury, Msgn1 and TBX6 and then downregulate brachyury to adopt a PSM state (Msgn1+/Tbx6+). Also, in the dot plot Fig 1g the authors should add the expression of Msgn1.

We have modified Fig. 1c, clarifying that MPCs are *T-Bra^{pos}*, *Msgn1^{pos}* and *Tbx6^{pos}* and that PSM cells are instead *Msgn1^{pos}*, *Tbx6^{pos}* and *Meox1^{pos}*. We also added *Msgn1* in the dot plot of Fig. 1g as requested.

Figure 3C: From the heatmap it is not clear if there are still NMP cells present at 36h and 48h of differentiation. It seems that the number of NMP cells using the described protocol is very low at 48h in wild type cells (less than 5% in Fig 3i). The authors should look more carefully if the cells that express posterior Hox genes are NMPs at 36 and 48 hours or pre neural tube cells that co-express Sox2/Nkx1.2.

Based on the remarks of the reviewer, we have revisited the cell identity of the NMP cells at 36 h and 48 h. The NMPs at 36 h and 48 h indeed possess also a pre-neural signature. Whether these cells are committed towards the neural lineage or if they retain mesodermal potential is an interesting aspect that we will investigate in our future projects. In the revised manuscript, we discussed the possibility that 36 h and 48 h NMPs are pre-neural fated cells and add this nomenclature to Fig. 3.

REVIEWERS' COMMENTS

Reviewer #1 (Remarks to the Author):

The authors have improved the manuscript with text changes and additional experiments. All my comments have been adequately answered.

Reviewer #2 (Remarks to the Author):

The authors have done a great job addressing my comments and I recommend this version of the manuscript for publication.

Hugo Parker